# Noninvasive Neuromodulation in Parkinson’s Disease: Insights from Animal Models

**DOI:** 10.3390/jcm12175448

**Published:** 2023-08-22

**Authors:** Katherine Muksuris, David M. Scarisbrick, James J. Mahoney, Mariya V. Cherkasova

**Affiliations:** 1Department of Psychology, West Virginia University, Morgantown, WV 26506, USA; 2Department of Behavioral Medicine and Psychiatry, Rockefeller Neuroscience Institute, West Virginia University, Morgantown, WV 26506, USA; 3Department of Neuroscience, Rockefeller Neuroscience Institute, West Virginia University, Morgantown, WV 26506, USA

**Keywords:** neurodegeneration, Parkinson’s Disease, animal models, noninvasive neuromodulation, transcranial magnetic stimulation, transcranial direct current stimulation, electroconvulsive stimulation, focused ultrasound

## Abstract

The mainstay treatments for Parkinson’s Disease (PD) have been limited to pharmacotherapy and deep brain stimulation. While these interventions are helpful, a new wave of research is investigating noninvasive neuromodulation methods as potential treatments. Some promising avenues have included transcranial magnetic stimulation (TMS), transcranial direct current stimulation (tDCS), electroconvulsive therapy (ECT), and focused ultrasound (FUS). While these methods are being tested in PD patients, investigations in animal models of PD have sought to elucidate their therapeutic mechanisms. In this rapid review, we assess the available animal literature on these noninvasive techniques and discuss the possible mechanisms mediating their therapeutic effects based on these findings.

## 1. Introduction

Parkinson’s Disease (PD) is a debilitating movement disorder usually diagnosed in individuals over 60 years of age. Its core symptoms are bradykinesia, rigidity, and impaired motor control in the limbs, as evidenced by tremors in the hands and a shuffling gait. However, the importance of cognitive, mood, and digestive symptoms associated with the disorder is being increasingly appreciated [1]. Indeed, non-motor symptoms are often more debilitating for patients and caregivers [2]. Considering the life expectancy in most industrialized nations [3], individuals diagnosed with PD may have many remaining years of life, which underscores the importance of effective treatment and management of this disease.

The approach to the treatment of PD has remained relatively stable for many years. The mainstay treatment is dopamine replacement therapy. Deep brain stimulation (DBS) is also an effective treatment but requires invasive surgery. Non-invasive neuromodulation methods, such as transcranial magnetic stimulation (TMS), transcranial direct current stimulation (tDCS), and focused ultrasound (FUS), are currently being explored as potential treatment options in PD with early evidence of efficacy in human patients [4,5,6]. Studies in animal models have helped elucidate the possible neurobiological mechanisms of these therapeutic effects, outlining future directions for human research. In this rapid review, we surveyed the current literature on major forms of non-invasive brain modulation in animal models of PD: repetitive transcranial stimulation (rTMS), transcranial direct current stimulation (tDCS), electroconvulsive therapy (ECT), and focused ultrasound (FUS). Because this review focuses on brain modulation methods, other forms of non-invasive stimulation that have been considered for the treatment of PD, such as transcutaneous nerve stimulation or electro-acupuncture, are beyond the scope of this review.

## 2. Materials and Methods

In this rapid review, we performed searches using the advanced search function in the National Institute of Health’s Pubmed website for keywords related to non-invasive methods targeting PD in animal models. We set our range of publication dates from 1980 to 2023, but no study we found was older than 1992. The keywords varied slightly for each category of non-invasive method. All searches included “Parkinson’s Disease” AND “animal model”, with the following terms also included: TMS—“AND transcranial magnetic stimulation”, “AND TMS”; tDCS—“AND transcranial direct current stimulation”, “AND transcranial alternating current stimulation”, “AND tDCS”, “AND tACS”; ECT—“AND electroconvulsive therapy”, “AND ECT”; and FUS—“AND focused ultrasound”, “AND transcranial ultrasound”, “AND FUS”, “AND TUS”, “AND LIFU”. These initial searches yielded 91 articles. Additional more inclusive searches using keywords “Parkinson’s disease” combined with “AND transcranial magnetic stimulation”, “AND TMS”; “AND transcranial direct current stimulation”, “AND tDCS”, “AND tACS”; “AND electroconvulsive therapy”, “AND ECT”; “AND focused ultrasound”, “AND FUS”, “AND TUS”, “AND LIFU” and with the filters set to “Other Animals” (i.e., excluding humans) yielded another 200 articles, some of which were duplicates of those resulting from the initial searches. Criteria for exclusion consisted of human studies, studies only using in vitro methods, review articles, meta-analyses, conference papers or presentations, and animal studies investigating neurodegenerative disorders generally instead of focusing on PD specifically. After reviewing the 291 articles for content in their abstracts (performed by the first author) and applying the exclusion criteria, we excluded 243 articles that did not meet the criteria for inclusion or were duplicates. This screening yielded 36 articles included in this review. Additional eligible articles were discovered in the references of the articles found by the literature search: two in the TMS category, eleven in the FUS category, and one in the ECT category. This made for a total of 50 articles included in this review.

## 3. Results

In the following sections, we will summarize studies by category of non-invasive neuromodulatory technique.

### 3.1. Transcranial Magnetic Stimulation (TMS)

#### 3.1.1. Background

TMS generates a high-intensity magnetic field by discharging an electrical current through a magnetic coil. This creates a magnetic field perpendicular to the coil that passes through the skull without attenuation. This induces an electrical field perpendicular to the magnetic field. The induced electrical field modulates the ongoing activity of neurons at the site of stimulation as well as in neural circuits encompassing the stimulation site. Single TMS pulses cause the temporally restricted modulation of neural activity and are used experimentally to study regional brain function. TMS pulses applied repetitively can produce lasting changes in neuronal excitability by inducing long-term potentiation (LTP)- or long-term depression (LTD)-like plasticity in stimulated regions and related circuits [7]. Owing to these lasting effects, repetitive transcranial magnetic stimulation (rTMS) has been used therapeutically in several disorders, most notably major depressive disorder and obsessive-compulsive disorder. The neuromodulatory effects of rTMS differ by frequency, with low-frequency (lf) rTMS causing a decrease in cortical excitability and high-frequency (hf) rTMS causing an increase in cortical excitability [8].

A considerable body of work has examined rTMS as a potential treatment for motor and cognitive symptoms of PD, with several meta-analyses and reviews on this topic providing evidence of efficacy in treatment of motor, depressive, and cognitive symptoms [9], as shown in Table 1. A number of mechanisms have been proposed to underlie the therapeutic effects of rTMS in PD based on both clinical and preclinical literature, including the induction of plasticity in the stimulated cortical areas and related circuits; a release of dopamine, which may modulate synaptic activity and plasticity; and molecular mechanisms bringing about neuroprotective effects [10,11]. Studies in animal models of PD make an important contribution to elucidating these mechanisms, especially those occurring at cellular and molecular levels.

#### 3.1.2. Animal Studies

The neural mechanisms underlying the therapeutic effects of TMS in PD were initially studied in wild-type (WT) rats prior to being investigated in PD models. We note two studies that investigated the effects of TMS on dopamine release [21,22]. Both studies found increases in striatal dopamine following hf-rTMS administration to the PFC, corroborating findings of Strafella et al. [23] in healthy human volunteers without PD. Notably, Kanno, et al. found that the acute administration of the 1-s trains of rTMS at 60% of maximum stimulator output achieved the greatest increases in dopamine compared to both 20% and 80% of stimulator output, indicating an inverted U-shape relationship between TMS intensity and striatal dopamine release [22].

A summary of TMS studies in animal models of PD is given in Table 2. All but three studies used a unilateral chemical lesioning model of PD with 6- hydroxydopamine (6-OHDA) [24,25,26,27,28,29,30,31,32] or lactacystin [33] in rats. Two of the remaining studies used bilateral lesioning via systemic injection of the toxin 1-methyl-4-phenyl-1,2,3,6-tetrahydropyridine (MPTP) in mice [34,35], and one additional study used unilateral 6-OHDA lesioning in mice [36]. Some of the studies used intermittent theta burst stimulation (iTBS), which is a form of excitatory hf-rTMS [26,27,28,29,30], while others utilized non-theta-burst low-frequency (inhibitory) or high-frequency (excitatory) rTMS [24,25,31,32,33]. The positioning of the TMS coil was typically over the dorsal surface of the animal’s head but was not neuroanatomically specific. However, there was a degree of functional specificity for the motor cortex, as the stimulation was reported by some of the studies to produce motor responses in limb muscles. The two iTBS studies achieved greater functional specificity, optimizing the positioning of the coil for eliciting motor-evoked potentials (MEPs) from the contralesional forelimb.

*Behavioral and Neuroprotective Effects*. The studies that examined the therapeutic effects of rTMS in the 6-OHDA rat model all found amelioration of toxin-induced motor disturbances by the chronic (several weeks, see Table 2) administration of rTMS or iTBS compared to sham control. Motor function was most frequently evaluated using the rotational behavior test, and all studies using this test reported improvements. The rotational behavior test assesses apomorphine- or amphetamine-induced rotational behavior in unilaterally lesioned animals, manifesting as vigorously turning in circles in the direction of the lesioned side. A slowing-down of these turns indicates the effectiveness of an intervention (in this case, rTMS) [37]. Other types of motor improvements were also noted by several of the studies (Table 2), as was improvement in memory performance and reductions in anxiety-like and depressive-like behavior [29]. All of these studies also reported that relative to sham treatment, chronic rTMS or iTBS produced an attenuation of the loss of tyrosine hydroxylase positive (TH+)—i.e., dopamine-producing—neurons in the substantia nigra (SN) on the lesioned side. One of the studies reported that the attenuation of SN cell loss was correlated with improvements in motor symptoms as measured by the rotational test [26]. This suggests that neuroprotection mediates the motor function benefits of rTMS.

The studies in mice using the MPTP model yielded findings consistent with those of the rat models, reporting rTMS-induced improvements in motor [34,35] and memory performance [35], an increase in neuronal preservation in the SN and dopamine in the striatum [34,35], decreases in inflammatory markers TNFα and interlukin-6 (IL-6) [35], and increases in NFs in the SN [34]. One of the studies [35] also showed that the rTMS-induced downregulation of microRNA miR-195a-5p (a putative biomarker for PD) with a resultant upregulation of cyclic AMP-response element-binding protein (CREB) may be a mechanism for the observed neuroprotection. The behavioral and neuroprotective benefits were observed with lf- [34,35] and hf-rTMS [35]. Another study in mice [36] using 6-OHDA rather than MPTP for lesioning investigated the effects of both lf- and hf-rTMS on amyloid pathology as a putative biomarker of PD [38]. In addition to improving memory performance and attenuating SN neuronal loss caused by 6-OHDA, lf- and hf-rTMS increased the levels of amyloid β_1–42_ (Aβ_1–42_) in the cerebrospinal fluid (CSF) while decreasing levels in the brain. This suggests that one of the therapeutic effects of rTMS may be the facilitation of Aβ_1–42_ clearance.

*Plasticity*. Studies have also pointed to synaptic plasticity as a possible therapeutic mechanism of iTBS. Ghiglieri, et al. (2012) [28] investigated the acute effects of iTBS on cortico-striatal plasticity and found a moderate restoration of LTD in cortico-striatal slices, which was lost following 6-OHDA lesions. The restoration of LTD was associated with an enhanced excitability of striatal neuronal populations. Using the same iTBS protocol, Cacace, et al. (2017) [30] investigated its acute effects on motor function, plasticity, and inflammation and reported the amelioration of gait disturbances and akinesia, as well as increased levels of striatal dopamine. Neither LTP nor LTD could be induced in striatal projection neurons of sham-treated 6-OHDA animals. iTBS normalized both LTP and LTD, and the rescue of plasticity was accompanied by c-fos activation in striatal spiny neurons. Finally, Jovanovic, et al. [29] reported that iTBS altered the composition of NMDA receptor subunits in the striatum, reversing the 6-OHDA-induced pathogenic changes to subunit composition. The change to subunits included an increased expression of the GluN2A subunit, which would be expected to facilitate synaptic plasticity. Together, these findings suggest that iTBS is capable of restoring deficient synaptic plasticity in Parkinsonian animals, helping normalize the basal ganglia circuit function.

*Anti-inflammatory Effects*. Several studies have reported on the anti-inflammatory effects of rTMS. Two studies examined the effects of lf-rTMS on inflammatory markers. Using a lactacystin-induced model of PD, Ba, et al. [33] found that 4 weeks of lf-rTMS administration reduced the levels of the pro-apoptotic enzyme caspace-3 and inflammatory markers tumor necrosis factor-α (TNFα), and cyclooxygenase-2 (COX-2). Yang, et al. [24] likewise found decreased TNFα and COX-2 levels with the same rTMS protocol in a 6-OHDA rat model. Cacace, et al. [30] found that their iTBS protocol significantly reduced astrogliosis and microglial activation produced by the 6-OHDA lesion in the ipsilesional striatum. Investigating the endocannabinoid system’s role in neuroinflammation in PD, Kang, et al. [31] found that hf-rTMS (but not lf-rTMS) downregulated the cannabinoid CB2 receptor in reactive astrocytes and its ligands in the ventral midbrain of 6-OHDA- and lipopolysaccharide (LPS)-lesioned rats while reducing inflammatory markers and preventing cell loss in the SN, as well as improving rotational performance. The selective agonism of the CB2 receptor blocked the rTMS-induced suppression of astrocyte activity. The CB2 receptor has been implicated in neuroinflammatory mechanisms of PD [39].

*Neurotrophic Factors*. rTMS-induced increases in neurotrophic factors (NFs) have also been reported and proposed to mediate neuroprotective effects of rTMS. Lee, et al. [25] found that rTMS-treated 6-OHDA rats had higher levels of NFs in the SN, including brain-derived neurotrophic factor (BDNF) and glial-derived neurotrophic factor (GDNF) amongst other trophic factors. Jovanovic, et al. [29] also reported increased BDNF in the striatum following chronic iTBS, and Ba et al. [32] reported increased GDNF in the SN following lf-rTMS.

*Levodopa-induced dyskinesias*. In addition to investigating the effects of rTMS on the cardinal symptoms of PD in animal models, one study investigated the effects lf-rTMS on levodopa-induced dyskinesia (LID) in a 6-OHDA rat model of PD. Ba, et al. [32] found decreases in abnormal involuntary movements, as well as reductions in levodopa-induced dopamine fluctuations in the striatum of 6-OHDA-lesioned rats that had received lf-rTMS in addition to levodopa. rTMS also attenuated the loss of TH+ cells and produced an increase in GDNF in the SN. As a possible mechanism, rTMS decreased the levels of the NR2B NMDA receptor subunit tyrosine phosphorylation in the ipsilesional striatum, which has been implicated in LID [40,41].

*Cortical Excitability*. Lastly, one of the iTBS studies evaluated the effect of 6-OHDA lesioning on the acute potentiation of motor cortical excitability by iTBS rather than evaluating the therapeutic effects of iTBS in this PD model [26]. A loss of enhancement of MEPs by iTBS was reported in the lesioned animals (relative to intact controls). The degree of MEP enhancement was inversely associated with the loss of dopaminergic neurons in the SN and their fibers in the striatum, as well as with the degree of motor impairment. These findings suggest that 6-OHDA lesioning impairs motor cortical plasticity, which is consistent with reports in human patients [42] and with the reported loss of LTP and LTD in cortico-striatal neurons of 6-OHDA-lesioned rats [28,30].

In summary, studies in animal models of PD have demonstrated that lf-rTMS, hf-rTMS, and iTBS attenuate the PD-related motor deficits and concomitantly reduce the loss of dopamine-producing SN neurons. The effects of hf-rTMS were superior to those of lf-rTMS for some outcomes. Notably, the effects of rTMS on the cognitive and affective symptoms of PD in animal models have received little attention. Considering that rTMS produces changes in cortical excitability of all stimulated regions, effects beyond those on motor function are expected. Therefore, evaluating TMS effects on cognition and affect may be a promising future research direction for studies in PD animal models.

Two findings emerging prominently from this still limited literature are rTMS-induced decreases in inflammatory markers and increases in NFs. None of the studies thus far have causally linked the anti-neuroinflammatory effects of rTMS to NF increases, which could be addressed by future research. Because lesioning itself causes inflammation, utilizing genetic models of PD to study the effects of rTMS on inflammation would strengthen the evidence of anti-inflammatory effects.

Neuroinflammation is recognized as playing a role in the pathophysiology of PD with several reviews on the topic [43,44,45]. However, the human literature has rarely considered TMS as a potential anti-neuroinflammatory treatment. One study reported that rTMS reduced proinflammatory cytokines in serum of PD patients [46]. Although neuroinflammation is not easily studied in humans, such studies are possible using positron emission tomography (PET) with tracers targeting the proxy markers of neuroinflammation. The most commonly used have been tracers targeting the translocator protein (TSPO), which is upregulated in activated microglia and is therefore considered to be a proxy measure of microglial activation [47]. There is a growing body of TSPO PET imaging literature in human PD patients [48], and tracers targeting other proxies of neuroinflammation are being tested in PD (e.g., [49]). Imaging the effect of rTMS on neuroinflammatory markers in human PD patients using such tracers may be a promising future direction. The link between NFs and the alleviation of PD symptoms is well characterized, and BDNF is now being considered as a potential treatment for PD [50]. However, the idea that TMS may be able to increase NFs is novel and worth further investigation.

Finally, although the induction of plasticity is a well-accepted therapeutic mechanism of rTMS, the animal studies reviewed here have provided considerable insight into the molecular mechanisms pertaining to the treatment of PD.

**Table 2 jcm-12-05448-t002:** Specifications of TMS studies. SN = substantia nigra; 6-OHDA = 6-hydroxydopamine; MPTP = 1-methyl-4-phenyl-1,2,3,6-tetrahydropyridine; LPS = lipopolysaccharide; MT = motor threshold; MEP = motor-evoked potential (induced by single TMS pulses); TNFα = tumor necrosis factor-α; COX-2 = cyclooxygenase-2; BDNF = brain-derived neurotrophic factor; GDNF = glial-derived neurotrophic factor; PDNF = platelet-derived neurotrophic factor; VEGF = vascular endothelial growth factor; NMDA = N-methyl-D-aspartate; CSF = cerebrospinal fluid; miR-195a5p = micro RNA-195a5p; CREB = cyclic AMP-response element-binding protein; Aβ = amyloid β; LTP = long-term potentiation; LTD = long-term depression; AEA = anandamide; 2-AG = 2-arachidonoylglycerol.

Study	PD Model	Sham Control	TMS Parameters	Outcome Measures and Findings
Ba, et al. (2017) [33]	Lactacystin-induced (rat)	Sham with stimulator sound but 2 cm above head No sham microinjection	0.5 Hz rTMS, once a day for 4 weeks Figure-8 coil Intensity: 250 V/m (100% MT)	Improvements in rotational test Increased TH+ neurons counts in SN Increased dopamine levels in striatum Reduced cleaved caspace-3, TNFα, and COX-2 in SN
Yang, et al. (2010) [24]	6-OHDA (rat)	Sham with stimulator sound but 1 cm above head No sham lesioning	0.5 Hz rTMS, once a day for 4 weeks Figure-8 coil Intensity: 250 V/m (100% MT)	Improvements in rotational test Increased TH+ levels in SN Decreased levels of COX-2 and TNFα in SN
Ghiglieri, et al. (2012) [28]	6-OHDA (rat)	No sham stimulation group Sham lesioning group	iTBS: 10 trains of 50 Hz bursts repeated at 5 Hz, at 10 s intervals delivered one time Figure-8 coil Intensity: 30% machine output	Restoration of LTD in cortico-striatal slices Increase in excitability of striatal neuronal populations measured as field excitatory postsynaptic potentials
Lee, et al. (2013) [25]	6-OHDA (rat)	No sham control of stimulation or lesioning	10 Hz rTMS once a day for 4 weeks Circular coil Intensity: 100% stimulator output	Improvements in rotational test Improvement in treadmill locomotion Increased TH+ neuron counts in SN Increased dopaminergic projections to striatum Increased levels of BDNF, GDNF, PDNF, and VEGF in SN
Dong, et al. (2015) [34]	MPTP (mouse)	Sham TMS with stimulator sound but 10 cm above head Sham lesioning group	5 trains of 1 Hz rTMS once a day for 14 days Circular coil Intensity: 1 Tesla	Improvements in rotarod performance No improvements in general locomotor activity Decrease in resting motor threshold Increased levels of dopamine and its metabolites in striatum Increased TH+, GDNF, and BDNF staining in SN
Hsieh, et al., 2015 [26]	6-OHDA (rat)	Sham with stimulator sound but coil 8 cm laterally and above head No sham lesioning	iTBS: triplets of pulses at 50 Hz repeated every 200 ms 2 s of TBS repeated every 10 s for 20 repetitions, single administration (acute) Intensity: 80% resting MT Coil positioned over dorsal scalp to optimally elicit MEPs in contralesional forelimb	Loss of enhancement of MEPs by iTBS in 6-OHDA-lesioned (relative to intact) rats iTBS-induced enhancement in MEPs negatively correlated with TH+ cell count in SN and TH+ fibers in striatum iTBS-induced enhancement in MEPs negatively correlated with the number of rotations on the rotational test
Cacace, et al. (2017) [30]	6-OHDA (rat)	No sham stimulation group Sham lesioning group	iTBS: 10 trains of 50 Hz bursts repeated at 5 Hz, at 10 s intervals delivered one time Figure-8 coil Intensity: 30% machine output	Improved gate and reduced limb akinesia at 80 min post-iTBS Increased striatal dopamine levels at 80 min post-iTBS Rescue of LTD observed at 20 min post-iTBS Both LTP and LTD rescued at 80 min post-iTBS as well as increased c-fos expression in striatal spiny neurons at 80 min post-iTBS Reduced astrogliosis and microglial activation in ipsilesional striatum at 80 min post-iTBS
Ba, et al. (2019) [36]	6-OHDA (mouse)	No sham stimulation group Sham lesioning group	Two sessions of 1000 pulses in 10 trains every day for 14 days (1 Hz and 10 Hz) Coil type unspecified Intensity: 1.26 Tesla	Improvement in Morris water maze performance Attenuation of neuronal loss in SN caused by 6-OHDA Increased HT and BDNF in SN Increased levels of Aβ_1–42_ in CSF and decreased levels in whole brain
Hsieh, et al. (2021) [27]	6-OHDA (rat)	Sham with stimulator sound but coil 8 cm laterally and above head No sham lesioning	iTBS: triplets of pulses at 50 Hz repeated every 200 ms 2 s of TBS repeated every 10 s for 20 repetitions, once a day for 4 weeks Figure-8 coil Intensity: 80% resting MT Coil positioned over dorsal scalp to optimally elicit MEPs in contralesional forelimb	Improvement in rotational test Improvement in bar test for akinesia Improvement in gait characteristics Increased TH+ neuron count in the SN Increased TH+ fibers in striatum
Ba, et al. (2016) [32]	6-OHDA + levodopa (rat)	Sham stimulation (unspecified) No sham lesioning group	Monophasic 0.5 Hz pulses, 500 pulses once a day for 3 weeks (co-administered with levodopa (25 mg/kg) twice a day) Figure-8 coil Intensity: 250 V/m	Attenuation of abnormal involuntary movements Attenuation of TH+ neuron loss in lesioned SN Reduced levodopa-induced dopamine fluctuations in ipsilesional striatum Increased levels of GDNF in lesioned SN Attenuation of NR2 B tyrosine phosphorylation Attenuation of interactions of NR2B and Fyn tyrosine kinase
Sun, et al. (2023) [35]	MPTP (mouse)	No sham stimulation group Sham lesioning group	1000 pulses in 10 trains daily for 3 weeks (1 Hz or 10 Hz) Coil type unspecified Intensity: 1.3 Tesla	Improvement on rotarod and Morris water maze tests Reduction in MPTP-induced neuronal loss in SN Increased levels of BDNF and TH in SN Decreased TNFα and IL-6 in CSF Downregulation of miR-195a5p and upregulation of CREB
Kang, et al. (2022) [31]	6-OHDA or LPS (rat)	Sham stimulation with coil held 10 cm above head No sham lesioning	500 pulses per day for 4 weeks (1 Hz and 10 Hz) Circular coil Intensity: 20% maximum stimulator output	Hf-rTMS improved performance on the rotational test Hf-rTMS attenuated 6-OHDA-induced TH+ cell loss in SN Hf-rTMS prevented increases in TNFα, IL-β, and IL-6 in the SN caused by 6-OHDA Hf-rTMS reduced CB2R in reactive astrocytes and its ligands AEA and 2-AG in ventral midbrain in 6-OHDA and LPS models
Jovanovic, et al. (2023) [29]	6-OHDA (rat)	Sham with stimulator sound Sham lesioning group	20 trains of 10 bursts at 50 Hz once a day for 21 days (iTBS) Figure-8 coil Intensity: 35% machine output	Improvement on the rotarod test and cylinder test Improvement on object recognition Attenuation of anxiety and depressive-like behaviors Reduced TH+ cell loss in SNc and reduced loss of TH+ fibers in striatum Increases in dopamine in striatum on lesioned side Alteration of NMDA receptor subunit composition in striatum Increased BDNF in striatum

#### 3.1.3. Limitations

One limitation that was noted in most studies surveyed was the lack of regional specificity of TMS in the rodent models. Physical constraints on the size of the TMS coil make it difficult to target a particular brain area in rats and mice. This limits the translatability of the findings to human patients, who receive TMS to focal brain regions, as well as precluding the study of TMS effects at the circuit level. Another limitation to note is the stress experienced by the animals during the TMS procedure [22] since they are restrained and awake in most studies. Stress has been shown to increase dopamine release in the striatum [51] and could therefore be a confounding factor, although sham controls experienced similar levels of stress. It should be noted that not all studies reviewed here used sham control for rTMS. Finally, the studies in rodent models of PD have rarely evaluated the possible interactions between rTMS and dopamine replacement therapy. The literature in human patients has suggested that the effects of rTMS may be modulated by pharmacotherapy [47,48].

### 3.2. Transcranial Direct Current Stimulation (tDCS) and Transcranial Alternating Current Stimulation (tACS)

#### 3.2.1. Background

Although several forms of electrical brain stimulation have long been practiced as a therapy for psychiatric disorders, transcranial direct current stimulation (tDCS) has become increasingly widespread in recent years, transcending medical and scientific use. Communities of individuals now exist that practice do-it-yourself tDCS for self-improvement [52]. tDCS has emerged as a promising treatment for several psychiatric and neurological disorders, including PD [53,54,55,56,57]. The application of a constant direct current induces a subthreshold shift in resting membrane potentials of cortical neurons at the stimulated site [58]. Depending on the direction of current flow in relation to axonal orientation, this can be either a depolarizing or a hyperpolarizing effect. Anodal tDCS has been found to increase cortical excitability, whereas cathodal tDCS decreases cortical excitability [59]. Similar to rTMS, the lasting effects of tDCS are attributed to synaptic plasticity [58], although tDCS may potentiate concurrently ongoing plasticity rather than generate it in weakly active synapses that are not already undergoing plasticity [60]. The most common targets of tDCS for PD have been the dorsolateral prefrontal cortex (DLPFC) and the primary motor cortex (M1). One of the first studies to utilize tDCS in PD patients with concurrent depression found that anodal tDCS over the DLPFC improved performance on executive function tasks compared to sham tDCS (same stimulator delivering very low current) [56]. However, application of anodal tDCS to M1 has not generally produced improvement in motor symptoms according to a recent meta-analysis [61] though some individuals studies, (e.g., Fregni et al. [62]) have reported the amelioration of motor symptoms. In addition to tDCS, a variation on it called transcranial alternating current stimulation (tACS) has also been investigated in PD patients. tACS uses an oscillatory pattern of electrical current that is known to entrain the oscillatory activity of resonant neurons in the stimulated regions. This has been shown to modulate/reduce bradykinesia, normalize oscillations in M1 and other regions, and improve cognitive performance [63,64]. Furthermore, tACS over the motor cortex has been successfully used to accomplish the phase cancellation of tremor rhythm, which produced nearly a 50% reduction in tremor amplitude [65]. Studies in animal models of PD have contributed to elucidating the cellular and molecular mechanisms of the therapeutic effects of tDCS and tACS.

#### 3.2.2. Animal Models

The study of the neural mechanisms underlying the therapeutic effects of tDCS in PD started with the application of tDCS to WT rats to assess the effects on dopamine release in the striatum and striatal activity after stimulating the frontal cortex [66,67]. Tanaka, et al. [66] found that after cathodal (but not anodal) tDCS to frontal regions of the cortex, extracellular dopamine levels in the striatum increased relative to sham tDCS (with very low current). In contrast, Takano, et al. [67] found that anodal tDCS to the frontal cortex increased neural activity in the nucleus accumbens assessed by rodent fMRI. This group reasoned that because of lesioning concerns related to cathodal tDCS, anodal tDCS may be preferable, although stimulation intensity in Tanaka, et al. [66] fell within the safety guidelines put forth by Liebetanz, et al. [68]. Both of these initial studies, producing seemingly contradictory findings, only applied tDCS once for 10 min, and the animals were anesthetized during the stimulation.

*Behavioral and Neuroprotective Effects*. More recently, studies have addressed the effects of chronic tDCS (daily, for multiple days or weeks) over M1 in PD rodent models (for a summary of findings and tDCS parameters, see Table 3). All the studies using an MPTP bilateral lesioning model of PD in mice found that anodal tDCS over M1/frontal cortex was sufficient to improve motor performance, with tDCS effects being comparable to those of levodopa [69,70]. The majority of these studies also reported attenuated TH+ cell loss in the SN and/or the striatum, or increases in TH and dopamine levels in MPTP mice [69,70,71]. One study additionally reported a tDCS-induced decrease in alpha-synuclein in the SN pars compacta (SNc) [70], while another saw an increase in GDNF in the SNc in addition to the attenuation of TH+ cell loss and motor deficits following the application of tACS [72]. Studies in the 6-OHDA rat models analogously reported improvements in motor behavior following the application of anodal tDCS to the motor cortex, with one study additionally reporting an attenuation of TH+ cell loss in the SN [71]. Notably, tDCS in this study was initiated early—24 h following lesion.

In addition to improving motor function, tDCS may also improve anxiety-like behavior in rodent PD models. A study of cognitive- and neuropsychiatric-like symptoms in a 6-OHDA rat model of PD found that chronic anodal tDCS over M1 decreased anxiety-like behaviors in addition to ameliorating motor performance [71]. However, this study did not observe changes in depression-like behaviors or recognition memory. Utilizing a rotenone-induced rat model of PD, Kade, et al. [73] showed that *transcranial electrostimulation*—a variation on tACS developed by this group that is similar to cranial electrostimulation therapy—was able to reduce neurodegeneration in the SN and anxiety-like behavior when applied over the frontal cortex. It is important to note that this study did not include a sham electrical stimulation group, unlike the studies discussed thus far.

*Antioxidant Effects and Inhibition of Autophagy*. Furthermore, work in MPTP mice suggested that tDCS may decrease markers of oxidative stress and autophagy. Oxidative stress plays an important role in the degeneration of dopamine neurons in PD [74]. Lu, et al. [69] assessed the effects of chronic tDCS on markers of oxidative stress, including nonenzymatic malonaldehyde (MDA), antioxidant enzymes superoxide dismutase (SOD), and glutathione peroxidase (GSH-Px). MDA decreased while SOD and GSH-PX increased in the brains of MPTP mice that were treated with chronic anodal tDCS over the left frontal cortex [69]. These findings were correlated with improved performance on the rotarod test. Lee, et al. [70] showed that after 5 days of treatment, anodal tDCS over M1 inhibited autophagy, which has also been implicated in the pathophysiology of PD [75].

*Stem-cell Transplantation*. tDCS has also been investigated as a potential supporting therapy for embryonic cell transplantation into the striatum to alleviate PD-like symptoms in a 6-OHDA rat model [76]. The rats that received anodal tDCS over the motor cortex for 2 weeks after implantation showed higher levels of BDNF, enlarged graft volume, and improved motor function relative to rats that received sham tDCS. Conversely, cathodal tDCS was associated with a slight reduction in TH+ cell count in the striatum, although this was not significant.

*Non-human primates*. Finally, there has been one tDCS study in a nonhuman primate PD model [77]. This study used an aged (20 years old) advanced model of MPTP-induced parkinsonism [78]. Anodal tDCS applied over M1 produced a substantial improvement in walking ability and a reduction in tremor frequency and PD-like signs compared to sham tDCS. The acute effects of tDCS were transient (approximately 30 min). An improvement in PD signs was correlated with ‘accumulated stimulation’, representing a product of duration and intensity summed across treatment sessions. Post-mortem, tDCS of the right M1 was found to have produced increased c-fos expression in M1 and in the SNc on the treated side relative to the untreated side, suggesting increased neuronal activation.

To summarize, tDCS and tACS studies in animal models have generally corroborated the results of human studies demonstrating motor function improvement with anodal tDCS and tACS over frontal cortical regions. In addition, there is some initial evidence of the amelioration of anxiety-like behavior, although there has been no evidence of improvement in depressive-like behavior or cognitive performance thus far. tDCS has also been found to have neuroprotective effects, promoting the preservation of dopaminergic neurons in the SNc, decreasing levels of alpha-synuclein in these neurons [70,71], and increasing levels of NFs [72]. The study by Lu, et al. [69] additionally points to antioxidant effects of tDCS. These findings largely parallel findings reported by the TMS literature described earlier. Furthermore, some of this work yielded evidence of an adjunctive benefits of tDCS for stem cell transplantation, which is an intriguing future direction [76].

**Table 3 jcm-12-05448-t003:** Specifications of tDCS and tACS studies. TES = transcranial electrostimulation; 6-OHDA = 6-hydroxydopamine; MPTP = 1-methyl-4-phenyl-1,2,3,6-tetrahydropyridine; TH = tyrosine hydroxylase; MDA = malonaldehyde; SOD = superoxide dismutase; GSH-Px = glutathione peroxidase; SN = substantia nigra; SNc = substantia nigra pars compacta; M1 = primary motor cortex; PFC = prefrontal cortex; LC3 = microtubule-associated protein 1 light chain 3; p62 = sequestosome1/p62; PI3K = phosphoinositide 3-kinase; AMPK = AMP-activated protein kinase; mTOR = mechanistic target of rapamycin; ULK1 = unc-51-like kinase; BDNF = brain-derived neurotrophic factor; GDNF = glial-derived neurotrophic factor; ↑ = increase, ↓ = decrease.

Study	PD Model	Sham Control	tDCS Parameters	Outcome Measures and Findings
Li, et al. (2011) [79]	6-OHDA (rat)	Electrodes to skull but no stimulation No sham lesioning group	Anodal tDCS Current intensity: 80 μA and 40 μA 30 min daily for 9 days Stimulation site: M1	Decrease in ipsilesional bias No evidence of histological damage due to tDCS
Li, et al. (2015) [77]	MPTP (monkey)	Electrodes to skull but no stimulation No sham lesioning group	Anodal tDCS Current intensity:0–2.5 mA 0–60 min daily for 12 days Stimulation site: M1, PFC, left temporal lobe, right temporal lobe	Decrease in motor PD signs (scored on a scale) Decrease in tremor frequency Increased walking Increased c-fos-stained cells in M1 and SN
Lu, et al. (2015) [69]	MPTP (mouse)	Sham tDCS used but nature of sham unspecified No sham lesioning group	Anodal tDCS Current intensity: 0.2 mA 10 min daily for 3 weeks Stimulation site: frontal cortex	Increased time on rotarod Increased dopamine and TH ^†^ decreased MDA ^†^ increased SOD and GSH-Px ^†^ ^†^ in whole mouse brain
Winkler, et al. (2017) [76] *	6-OHDA (rat)	Electrodes to skull but no stimulation No sham lesioning group	Anodal tDCS Current intensity: 8 mA 20 min daily for 14 days Stimulation site: motor cortex	Reduction in rotational behavior Increased BDNF Enlarged graft volume in striatum
Lee, et al. (2018) [70] **	MPTP (mouse)	Sham tDCS used but nature of sham unspecified Vehicle injection group	Anodal tDCS Current intensity: 0.1 mA 30 min daily for 5 days Stimulation site: M1	Increased time on rotarod Increased TH and TH+ neuron count in SNc Decreased alpha-synuclein in SNc Lower levels of autophagy related proteins in SNc (LC3 and p62) Altered markers upstream of autophagy (↑ PI3K, ↑ mTOR, ↓ AMPK, ↓ ULK1) Increased levels of BDNF
Kade, et al. (2019) [73]	Rotenone (rat)	No sham control No sham lesioning group	TES, alternating current Stimulation intensity: 1.7 mA; frequency: 70 ± 2 Hz, 3.75 ± 0.25 ms pulses 30 min, 7 days Stimulation sites: frontal and mastoid	Decreased in motor PD signs (rated on a scale) relative to pre-TES baseline Decreased in anxiety behaviors in open field test relative to pre-TES baseline Decreased neurodegeneration in SNc relative to TES-untreated controls
Feng, et al. (2020) [71]	6-OHDA (rat)	Electrodes to skull but no stimulation No sham lesioning group	Anodal tDCS Stimulation intensity: 300 μA 20 min/day, 5 days/week for 4 weeks Stimulation site: M1	Improved motor performance Decreased anxiety-like behavior Increased TH+ cells in SN and fibers in striatum No improvement in rotational behavior No improvements in depressive-like behaviors No improvement in recognition memory
Lee, et al. (2022) [72]	MPTP (mouse)	Sham tDCS used but nature of sham unspecified No sham lesioning group	High-definition tACS Stimulation frequency: 89.1 μA/mm^2^; frequency: range 6–60 Hz; greatest effects at 20 Hz 20 min/day for 5 days Stimulation site: M1	Improved motor performance Increased TH+ cell count in SNc and striatum Less cleaved caspace-3 staining in SNc Increased GDNF in striatal parvalbumin+ neurons

* This study utilized tDCS to enhance dopaminergic cell transplantation. ** This study also performed in-vitro investigations, results not discussed in this review.

#### 3.2.3. Limitations

Compared to TMS, tDCS and tACS techniques have somewhat improved spatial resolution in small animal models because of the use of electrodes, which facilitates targeting specific cortical areas. Unlike TMS, tDCS in rodents necessitates the removal of the scalp which requires surgery, causing additional stress that is not associated with this treatment in humans and non-human primates [77]. Another limitation is the variation in stimulation protocols and parameters in the literature we found. Some studies, such as the initial WT animal studies [66,67] only applied tDCS acutely. Some others [69,70] administered tDCS while the animals were still being treated with MPTP to induce the PD-like phenotype and continued tDCS administration for weeks after MPTP treatment had ended. In contrast, Feng et al. [71] only started tDCS after PD lesioning had concluded and administered tDCS chronically for four weeks. Treatment duration is an important variable when considering the translational utility of these findings for clinical applications. In addition, few studies have addressed the effects of tDCS on the non-motor symptoms of PD, with the existing evidence not demonstrating an improvement in depressive-like behavior or cognition. Finally, tDCS studies have yet to evaluate interactions between tDCS and dopamine replacement therapy in PD animal models. Overall, the animal literature on tDCS positions it as a promising treatment that, in addition to modulating circuit activity to improve motor and non-motor symptoms, may have neuroprotective/regenerative effects. For some of these effects parallel those seen with rTMS, however, we note that at the present time, the rTMS animal model literature is more extensive. On the other hand, the cost-effectiveness of tDCS promises enhanced versatility in clinical use.

### 3.3. Electroconvulsive Therapy (ECT)

#### 3.3.1. Background

ECT has been used to treat psychiatric disorders since the early 1940s and is still in use today as an effective treatment option for those who suffer from refractory depression [80]. Aside from its use in psychiatry, ECT has also been shown to be effective for treatment of motor disturbances in PD and other movement disorders [81], although it is rarely used in clinical practice. The exact mechanisms underlying the therapeutic effects of this technique in PD are yet to be fully elucidated, which is why we turn to animal models (see Table 4 for a summary of studies).

#### 3.3.2. Animal Models

*Behavioral, Dopaminergic System, and Neuroprotective Effects.* The earliest ECT study (Garcia & Sotelo, 1993) examined the effect of ECT on the levels of striatal dopamine in the MPTP mouse model [82], finding no difference between the ECT-treated and -untreated mice. Following a considerable gap in the literature after this initial discouraging finding, two studies from a single group reported improvement in motor functioning following ECT in 6-OHDA-lesioned rats [83,84]. Strome, et al. [84] applied ECT (or sham: animals anesthetized, electrodes placed, but no current delivered, see Table 4) for 10 days after 6-OHDA lesioning and found an improvement in hind limb motor functioning in ECT-treated animals. This group also investigated the binding of dopamine receptors in the striatum using autoradiography and found significant increases in binding for D1 and D3 receptors but not D2 receptors in ECT-treated animals. Anastasía, et al. [83] found that the daily administration of ECT (relative to sham ECT) for 7 or 14 days concurrently with 6-OHDA lesioning reduced motor impairment and the loss of TH+ neurons in the SNc; ECT also increased the amount of GDNF in the striatum of both lesioned and un-lesioned rats. A subsequent study by the same group demonstrated that GDNF increases mediated the preservation of dopaminergic neurons in the SNc by ECT [85].

A study in MPTP-lesioned non-human primates found no significant changes in either dopamine D2/3 receptor or the dopamine transporter availability after ECT was delivered 6 times in 3 weeks [86]. However, there were significant increases in dopamine D1 receptor and the vesicular monoamine transporter (VMAT) availability. In un-lesioned animals, ECT increased dopamine transporter, D1 receptor, and VMAT binding, which is consistent with earlier findings in intact monkeys, and the effect of ECT on VMAT and D1 binding was less robust in the lesioned than in the un-lesioned striata. This suggests that while dopaminergic response to ECT may be weakened, it is to some degree preserved in MPTP-lesioned striatum.

*Stem Cell Transplantation*. Approximately a decade later, three additional studies evaluated ECT in the MPTP mouse model. Yang, et al. [87] combined ECT with the transplantation of mesenchymal stem cells (MSCs) in an MPTP mouse model of PD. ECT was delivered for 8 days following MSC transplantation. While they found that ECT and the MSCs on their own slightly increased dopamine concentrations in the whole brain, when combined, there was a significant increase in dopamine and TH levels [87]. This study also found improvements in motor behavior in the MSC + ECT group relative to animals that received either ECT or MSCs alone.

*Autophagy*. Huh, et al. [88] investigated the role of ECT in autophagy in the MPTP mouse model and found that after two weeks of ECT treatment (once a day, three times a week), mice showed an improvement in motor deficits; increased count of TH+ neurons in the SNc; and the normalization of LC3-II levels in the PFC, striatum, and midbrain. LC3-II is an autophagy marker that has been shown to be dysregulated in PD [89] and was found to be modulated by tDCS [17], as reported above.

In summary, relatively few studies have evaluated the effects of ECT in PD animal models. The extant studies have demonstrated an amelioration in motor deficits, the preservation of dopaminergic neurons in the SNc, the enhancement of dopaminergic function, and increases in NFs [83,84], paralleling findings with rTMS and tDCS. In addition, like tDCS, ECT may affect autophagy pathways and has shown promise as an adjunctive treatment for stem cell transplantation [87].

**Table 4 jcm-12-05448-t004:** Specifications of ECT studies. 6-OHDA = 6-hydroxydopamine; MPTP = 1-methyl-4-phenyl-1,2,3,6-tetrahydropyridine; SNc = substantia nigra pars compacta; PFC = prefrontal cortex; TH = tyrosine hydroxylase; GDNF = glial-derived neurotrophic factor; MSCs = mesenchymal stem cells; VMAT = vesicular monoamine transporters; LC3-II = microtubule-associated protein 1 light chain 3 conjugate; DAT = dopamine transporter; ↑ = increase, ↓ = decrease.

Study	PD Model	Sham Control	ECT Parameters	Outcome Measures
Garcia & Sotelo (1993) [82]	MPTP (mouse)	No sham stimulation group Sham lesioning group	Electrode placement unspecified 2 s pulses, 350 mV once a day for 5 days Shock intensity	No difference in dopamine levels in untreated vs. treated group
Anastasia, et al. (2007) [83]	6-OHDA (rat)	Electrodes placed but no current delivered Sham lesioning group	Corneal electrodes 0.2 s, 200 pulses/s, once per day for 1 or 3 days (acute) and then for 2 weeks Shock intensity: 40 mA	Decreased rotational behavior Attenuated TH+ neuronal loss in SNc Increased GDNF in SN and striatum
Strome, et al. (2007) [84]	6-OHDA (rat)	Electrodes placed but no current delivered No sham lesioning group	Earclip electrodes 5–9.9 s, 70 pulses/s once per day for 10 days Shock intensity: 80–99 mA	Improved hindlimb but not forelimb motor performance Increased D1 and D3 binding in dorsal and ventral striatum No change in D2 or VMAT binding
Anastasia, et al. (2011) [85]	6-OHDA (rat)	Electrodes placed but no current delivered Sham lesioning group	Supra-orbital electrodes 0.2 s, 200 pulses/s, once per day for 15 days Shock intensity: 40 mA	Increased GDNF expression in SNc but not striatum Increased TH+ cells count in SNc GDNF antibodies blocked the preservation of TH+ neurons in SNc Increased astrocyte count in anterior but not posterior SNc
Landau, et al. (2012) [86]	MPTP (monkey)	No sham stimulation group No sham lesioning group (bilateral vs. unilateral lesions vs. intact comparisons)	Electrodes applied to temples 0.5 s, 70 Hz, 6 times over 3 weeks Shock intensity: 0.9 A	No change in DAT binding in the lesioned striatum No change in D2/3 receptor binding in lesioned or un-lesioned striata Increases in VMAT2 and D1 receptor binding in lesioned striata
Yang, et al. (2020) [87] *	MPTP (mouse)	Animals anesthetized, electrodes placed but no current delivered Sham injection group	Earclip electrodes 1 s, 200 pulses/s, once per day for 8 days Shock intensity: 80 mA	Increased survival of MSCs Synergistic behavioral and neurobiological effects of MSC transplantation and ECT Decreased rotational behavior and improved stepping test performance Increased dopamine and TH levels in the whole brain Decreased proinflammatory cytokines
Huh, et al. (2023) [88]	MPTP (mouse)	Electrodes placed but no current delivered No sham lesioning group	Earclip electrodes 0.4 s, 60 pulses/s, 3 times a week for 2 weeks Shock intensity: 55 mA	Increased time on rotarod Increased TH+ cell count in SNc Normalization of autophagy markers (LC3-II ↓ in midbrain, ↑ in PFC)

* This study utilized ECT as a supplementary therapy to enhance transplantation of mesenchymal stem cells into a mouse model of PD.

#### 3.3.3. Limitations

Although the studies in animal models have clarified some of the mechanisms whereby ECT improves motor functioning, the literature is sparse. Furthermore, effects on non-motor PD symptoms and cognition have not been evaluated, the latter being especially important considering documented transient cognitive side effects resulting from ECT in humans [90]. Such side effects may be concerning in patients already experiencing cognitive impairment. The fact that the animals in most of the above studies were not under general anesthesia during ECT limits the translational utility of the findings. In humans, ECT is only performed under general anesthesia. Although important for seizure confirmation, the absence of anesthesia likely results in stress. Another variable to consider is the placement of the electrodes. Some studies [83,85] utilized corneal electrodes which stimulate the brain through the eyes, while the others utilized earclip [84,87,88] or temple [86] electrodes. Considering that ECT and general anesthesia used to deliver it in humans both come with risks, rTMS and tDCS may be preferable for use in clinical practice, as they offer similar benefits based on the animal literature reviewed.

### 3.4. Focused Ultrasound (FUS)

#### 3.4.1. Background

FUS relies on ultrasound waves guided to target focal regions in the body or the brain. Transcranial FUS has been studied and used for the treatment of brain disorders. High-intensity FUS (HIFU) has been used to produce focal brain lesions [4], whereas low-intensity FUS (LIFU) has been studied as a method of modulating neural activity [91,92] or of aiding drug delivery to the brain by focally and transiently disrupting the integrity of the blood–brain barrier (BBB) through the sonication of injected microbubbles (MBs) [93], (Figure 1). In contrast to the other non-invasive neuromodulation techniques reviewed above, FUS can directly target deep subcortical regions and has a superior spatial resolution on the order of millimeters [94,95].

In human PD patients, HIFU has been utilized to ablate the ventral intermediate nucleus of the thalamus, the subthalamic nucleus (STN), the internal segment of the globus pallidus (GPi), and the pallidothalamic tract [96,97]. These ablation treatments have been encouraging, demonstrating improvement of PD motor signs with relatively few and mild side effects [96]. Although LIFU has not yet been trialed as a form of neuromodulation for PD treatment, LIFU (without microbubbles/BBB disruption) has been tested in proof-of-concept studies in healthy volunteers, as well as initial studies in epilepsy and generalized anxiety disorder [92,96]. A trial in opioid use disorder (NCT04197921) is currently underway and initial results are promising [98]. The use of low-intensity FUS with microbubbles (FUS-MBs) to disrupt the BBB has received considerable attention in Alzheimer’s disease [99,100,101] and is an emerging application in PD with a potential for targeted non-invasive delivery of therapeutics to the brain and/or for enhancing the clearance of protein aggregates (Figure 1). Thus far, a phase-1 open-label trial has investigated the safety and feasibility of applying FUS-MBs in PD patients with dementia, demonstrating that FUS-MBs safely and reversibly opened the BBB in the parieto-occipito-temporal cortex (targeted pathological site), as evidenced by gadolinium enhancement on MRI in this region [102]. The same group subsequently reported the safe and successful opening of the BBB in the putamen and midbrain of three PD patients (as evidenced by uptake of a PET tracer that does not cross the BBB), along with focal delivery of adeno-associated virus serotype 9 vectors into PD-relevant brain regions of macaque monkeys [103]. Although this is only an emerging area of research in human PD patients, there are many animal studies that have investigated the underlying mechanisms of FUS-MBs and have advanced the joint use of this technique with delivery of therapeutics. In the following sections, we discuss the application of LIFU with and without the use of microbubbles.

#### 3.4.2. LIFU Animal Models

*Behavioral and Neuroprotective Effects.* Studies of LIFU in animal models of PD began rather recently, with earliest studies published in 2019 [104,105,106]. Two of the earliest studies from the same group utilized an MPTP mouse model of PD and applied LIFU to the basal ganglia nuclei (STN and GP) [105] or the motor cortex [106]. Both studies utilized a wearable ultrasound device in awake behaving animals, but LIFU parameters differed between the two studies, with [106] employing a lower stimulation frequency (Table 5). In both studies, MPTP mice treated with LIFU showed improved motor performance in comparison to animals that were given sham LIFU (animals handled, stimulator placed, but no ultrasound given; Table 5). C-fos expression was increased in the sonicated regions, suggesting that LIFU resulted in increased neural activity. Zhou, et al. [105] additionally found more TH+ neurons in the SN following LIFU, which was likely mediated by the suppression of apoptosis in the SN evidenced by a decrease in apoptosis markers. Zhou, et al. [106] reported that LIFU of the motor cortex increased the levels of antioxidant enzymes in the striatum. Neither study found any tissue damage in the brains of the mice that had received LIFU. Xu, et al. (2020) [107] corroborated the improvement of motor signs on behavioral tests and neuroprotection, reporting increased activity during the open field test and better performance on the pole test after 10 days of LIFU to the whole brain. TH+ neuron count in the SNc and striatal dopamine levels were increased in the MPTP mice that had received LIFU relative to those that did not.

*Anti-inflammatory Effects*. In a subsequent study, Zhou and colleagues further investigated the effects of LIFU on neuroinflammation in their MPTP model, with LIFU applied to the STN acutely (1 time for 30 min) [108]. In addition to replicating their earlier findings of motor performance improvement, the preservation of SN neurons, and increases in antioxidant enzymes, Zhou et al. (2021) [108] found reductions in the levels of proinflammatory cytokines and downstream inflammatory markers, as well as decreased microglial and astrocyte activity in the SN and striatum of MPTP mice that were treated with LIFU. There was also a decrease in alpha-synuclein in the SNc and the striatum [108]. Song, et al. [109] built upon this work by assessing neuroinflammation in a 6-OHDA rat model of PD after treatment with low-intensity pulsed ultrasound for 6 weeks targeting the lesion. Relative to no treatment, ultrasound prevented glial activation caused by 6-OHDA, decreased inflammatory markers, and produced an increase in GDNF (with no change in BDNF). Ultrasound also restored the integrity of the BBB in the lesioned and sonicated SN region.

Additional evidence of the anti-inflammatory and antioxidant effects of LIFU came from Dong, et al. (2021) [110], who utilized rodent functional magnetic resonance imaging (fMRI) to assess T2* relaxation time as a proxy measure of iron deposition in a 6-OHDA rat model. The study also examined the effects of LIFU on fractional anisotropy (FA) as a measure of neuronal integrity. Consistent with previous studies, there was an attenuation of the loss of TH+ neurons and an increase in GDNF of 6-OHDA rats treated with LIFU relative to untreated rats. After 5 weeks, the LIFU group showed higher T2* values, which was interpreted as resulting from decreased iron deposition into the surrounding tissues. Because iron deposition is a marker of oxidation and inflammation [111], LIFU was proposed to have reduced free radicals and inflammation through improving microcirculation in the targeted brain areas [110]. FA values were increased during the first week of treatment, which was interpreted as suggestive of neuroprotection, although this effect was reversed after 5 weeks of treatment, with FA values becoming smaller in treated relative to the untreated rats. Because MRI is non-invasive and routinely used in humans, the study raises the possibility of using these MRI-based measures in human patients to assess LIFU effects.

*Effects on Cognition*. Finally, one of the earlier studies utilized a variation of the focused ultrasound technique called transcranial magneto-acoustic stimulation (TMAS) in an MPTP mouse model of PD [104]. TMAS is performed by delivering focused ultrasound in a static magnetic field to further increase its spatial resolution [112]. Wang, et al. [104] delivered TMAS to the SN daily over 2 weeks and examined the effects on spatial learning and memory, hypothesizing that TMAS of the SN would produce the beneficial downstream modulation of basal ganglia—hippocampal circuits. TMAS improved performance on the Morris water maze and increased the dendritic spine densities in the dentate gyrus of the hippocampus, with a concomitant increase in proteins mediating plasticity (postsynaptic density protein 95 (PSD-95), BDNF, CREB, and protein kinase B).

**Table 5 jcm-12-05448-t005:** Specifications of LIFU studies. 6-OHDA = 6-hydroxydopamine; MPTP = 1-methyl-4-phenyl-1,2,3,6-tetrahydropyridine; PRF = pulse repetition frequency; TBD = tone burst duration; SD = sonication duration; ISI = interstimulus interval; SN = substantia nigra; SNc = substantia nigra pars compacta; STN = subthalamic nucleus; GP = globus pallidus; V1 = primary visual cortex; Bcl2 = B-cell lymphoma 2; Bax = bcl-2-like protein 4; Cyt C = cytochrome C; GSH-PX = glutathione peroxidase; SOD = superoxide dismutase; IL1β = interleukin-1β; NF-κBp65 = nuclear factor kappa-light-chain-enhancer of activated B cells p65; FA = fractional anisotropy; LTP = long-term potentiation; DG = dentate gyrus; PSD-95 = postsynaptic density protein 95; CREB = cyclic AMP response element-binding protein; BDNF = brain derived neurotrophic factor; PKB = protein kinase B; DAT = dopamine transporter; ↑ = increase, ↓ = decrease.

Study	PD Model	Sham Control	Ultrasound Parameters	Outcome Measures
Wang, et al. (2019) [104]	MPTP (mouse)	Mice placed in magnetic field, wore stimulator w/o stimulation delivered Saline injection	Transcranial magneto-acoustic stimulation (TMAS) (low-intensity) in static magnetic field of 0.17 T Target: SN Frequency: 1 MHz, PRF: 100 Hz, SD: 60 s Once a day for 2 weeks	Improved performance on Morris water maze Increased LTP and dendritic spine densities in hippocampus DG Increased plasticity-mediating proteins in hippocampus (PSD-95, BDNF, CREB, PKB) No damage to brain tissue
Zhou, et al. (2019) (2) [106]	MPTP (mouse)	Wore transducer w/o stimulation Saline injection	Low-frequency low-intensity pulsed US (LIPUS) Target: motor cortex Frequency: 800 kHz, PRF: 0.1 kHz, TBD: 1 ms, SD: 6 s, ISI: 10 s 40 min a day for 7 days	Improved performance on pole and more rearing activity in open field Increased c-fos expression in motor cortex Increased levels of antioxidant enzymes (SOD and GSH-PX) in striatum No damage to brain tissue
Zhou, et al. (2019) (1) [105]	MPTP (mouse)	Wore transducer w/o stimulation Saline injection	Low-intensity ultrasound deep brain stimulation Target: STN, GP Frequency: 3.8 MHz, PRF: 1 kHz, TBD: 0.5 ms, SD: 1 s, ISI: 4 s 30 min a day for 6 days	Improved performance on rotarod and pole tests (not open field test) Increased c-fos expression in STN and GP Increased TH+ neuron count in SNc Suppressed apoptosis in SN (↑ Bcl2/Bax, ↓ Cyt C release from mitochondria, ↓ cleaved caspase-3) No damage to brain tissue
Xu, et al. (2020) [107]	MPTP (mouse) and cell line (not discussed)	No sham LIFU Mice untreated with MPTP	Low-intensity ultrasound (0.1–0.3 W/cm^2^) Target: whole brain Frequency: 1 MHz 1–15 min a day, 1–10 days (dose-dependent effects evaluated)	Improved performance on open field test and pole test, scaling with stimulation intensity and duration Increased release of dopamine in striatum, scaling with stimulation intensity and duration Increased TH+ neurons count in SNc at highest stimulation intensity and duration No damage to brain tissue
Dong, et al. (2021) [110]	6-OHDA (rat)	No sham control for LIFU No sham lesioning	Low-intensity focused ultrasound Target: SN Frequency: 500 kHz, PRF: 1 kHz, TBD: 0.5 ms, SD: 400 ms, 10 min every day for 6 weeks	Increased FA in SN after 1 week of LIFU Decreased FA in SN after 5 weeks of LIFU Higher T2* values after 5 weeks of LIFU, less iron staining in SN Increased TH+ neuron count in SN Increased GDNF in SN Less iron in SN
Zhou, et al. (2021) [108]	MPTP (mouse)	Sham LIFU (see [36]), V1 LIFU Saline injection	Low-intensity focused ultrasound Target: STN Frequency: 3.8 MHz, PRF: 1 kHz, TBD: 0.5 ms, SD: 1 s, ISI: 4 s 30 min, once, the day after MPTP injection	Improved performance on rotarod and pole test Increased c-fos expression in STN (and V1 following V1 sonication) Decreased pro-inflammatory cytokines and downstream inflammatory signaling markers in SN and striatum Reduced microglial and astrocyte activation in SN and striatum Increased TH+ neuron count in SNc Decreased alpha-synuclein in SN and striatum Increased levels of antioxidant enzymes (SODs) in SN and striatum No damage to brain tissue
Song, et al. (2022) [109]	6-OHDA (rat)	No sham control for LIPUS No sham lesioning	Low-intensity pulsed US (LIPUS) Target: right hemisphere lesioned region Frequency:1 MHz, PRF: 1 Hz, SD: 5 min, ISI: 5 min 15 min, 5 days a week for 6 weeks	Reduced microglial and astrocyte activation in SNc Decreased inflammatory markers in SNc (IL1β, phosphorylated NF-κBp65) Increased GDNF but not BDNF in SNc Increase in DAT in SNc Restoration of BBB integrity in SNc (tight junction proteins increased) No damage to brain tissue

#### 3.4.3. FUS-MB Animal Models

Although NFs, such as BDNF and GDNF, have been identified as potentially promising therapeutics for neurodegenerative disorders, the BBB has presented an obstacle preventing their non-invasive delivery to the brain [113]. While the BBB’s permeability to BDNF is comparable to that of insulin [114], the BBB is impermeable to GDNF, necessitating intracerebral administration. A growing body of literature in animal models of neurodegenerative disorders is providing increasing support for the use of the FUS-MB BBB disruption technique as a non-invasive method of delivering therapeutics, such as NFs, to the brain, with encouraging findings regarding therapeutic efficacy.

*Delivery of GDNF*. One group of FUS studies in PD models yielded by our search have investigated whether FUS can be used to facilitate the non-invasive delivery of NFs or NF genes. This technique entails the systemic administration of a therapeutic (e.g., NF) and microbubbles, either in isolation or coupled together as a complex. FUS is then used to sonicate the focal brain area to which the therapeutic is targeted. FUS induces a rapid expansion and contraction of the MBs, which transiently disrupts the BBB, allowing for the delivery of the therapeutic to the parenchyma (Figure 1). The majority of these studies investigated the use of the FUS-MB technique for the delivery of GDNF through either protein or gene delivery to the SNc and/or striatum, including Fan, et al. [115], Lin, et al. [116], Mead, et al. [117], Yue, et al. [118], and Karakatsani, et al. [119] (see Table 6 for details of each study). All of these studies demonstrated significant improvements in motor function in their respective PD models, in some cases even reporting an almost complete restoration of motor function, with motor performance approaching un-lesioned levels, e.g., [117]. Every study also demonstrated increases in dopaminergic neurons and in the levels of dopamine (and/or other markers of dopaminergic signaling) in the targeted areas, with some reporting almost complete restoration to un-lesioned levels [116,117,118]. Yue, et al. [118] additionally found that behavioral improvements from GDNF delivery in FUS-MB-treated animals were associated with the increased expression of nuclear receptor-related factor 1 (Nurr1), which has been previously found to enhance the integrity of dopaminergic cells [120]. Some of the studies directly compared the therapeutic efficacy of injecting the NF genes together with, but uncoupled from, MBs versus injecting conjugated MB-gene complexes. Their findings suggested that conjugates produced superior therapeutic effects, although uncoupled delivery also produced significant benefits for some outcomes (as did FUB-MBs without gene delivery) [115,116,121]. One study aimed to increase the efficiency of GDNF gene delivery by designing a novel type of microbubbles loaded with polyethylenimine–superparamagnetic iron oxide plasmid DNA, used in conjunction with FUS and two-step magnetic navigation [122]. In cell culture, each component of this delivery system was reported to increase the transfection rate. In a genetic mouse PD model (MitoPark), the delivery of the GDNF gene to the SN using this system resulted in a 3.2-fold increase in the recovery of TH+ neurons and a 3.9-fold improvement in motor function relative to untreated PD mice. Although utilized in this study for GDNF gene delivery in the mouse model, this technology could be used for other types of gene delivery/therapy.

*Delivery of Other Neurotrophic Factors.* Besides GDNF, studies have investigated the FUS-MB-assisted delivery of other trophic factors, including neurturin (NTN) [119], BDNF [123], and fibroblast growth factor 20 (FGF20) [124]. Karakatsani et al. [119] investigated the protein delivery of NTN, as well as the gene delivery of GDNF, finding increases in dopamine neurons and dopamine levels in the targeted areas (SN and striatum). As NTN belongs to the GDNF family, the effects were similar to those observed after the FUS-MB-assisted gene delivery of GDNF, although they tended to be less pronounced with protein NTN delivery (some were not significant following a single NTN treatment and only observed after 3 deliveries). Ji, et al. [123] investigated FUS-MB-assisted delivery of BDNF in an MPTP mouse model using a unique approach; while the other studies injected both MBs and therapeutics intravenously, Ji et al. only injected MBs and administered BDNF intranasally. The intranasal administration of BDNF coupled with FUS-MBs increased the expression of TH in the SNc and striatum relative to the untreated hemisphere. FUS-MBs without intranasal BDNF administration produced no effect. At the behavioral level, amphetamine-induced rotational behavior towards the untreated side was observed, indicating increased dopaminergic activity on the treated side. Niu, et al. [125] used FUS-MBs to deliver FGF20, which is preferentially expressed in the SNc and has been found to enhance dopamine neuron differentiation from embryonic stem cells [126] and the survival of dopamine neurons in vitro [124]. Genetic variability in FGF20 is associated with risk for PD in humans [124], and in rodent PD models, FGF20 was found to be protective against the loss of dopamine neurons and concomitant motor dysfunction [124,127]. Therefore, FGF20 is a promising therapeutic for PD. Niu, et al. [125] utilized the FUS-assisted delivery of human recombinant FGF20 in proteoliposomes over two weeks to the brain of 6-OHDA-lesioned rats and found improvements on the rotational behavior test, as well as the alleviation of TH+ cell loss in the SN.

*Delivery of Antioxidants*. FUS-assisted delivery of other therapeutics has also been investigated in PD models. A study by Long, et al. [121] investigated the FUS-assisted delivery to the SN of nuclear factor E2-related factor 2 (Nrf2) gene coupled to nanomicrobubbles in the 6-OHDA rat model. Nrf2 is a transcription factor that activates an antioxidant response, which could be beneficial considering that oxidative stress plays an important role in the degeneration of dopamine neurons in PD [74]. In addition to increased Nrf2 expression, FUS-MBs also produced increases in TH and dopamine transporter (DAT) levels, as well as an increase in SOD and a decrease in reactive oxygen species in the SN.

*Delivery of Medicinal Herbs*. In addition, a series of studies examined the use of the FUS-MB technique to deliver medicinal herbs to the brain in PD mouse models. Two studies examined the FUS-assisted curcumin delivery to the striatum of MPTP-treated mice [128,129]. Curcumin may have neuroprotective effects in neurological disorders [130] and has also been shown to decrease alpha-synuclein aggregation in-vitro [131]. The two studies manufactured curcumin-loaded nanobubbles using different technologies and used FUS to sonicate these particles in order to facilitate curcumin delivery. Both studies reported motor improvements, while Zhang et al. [129] additionally reported increased dopamine and dopamine metabolites in the striatum and the preservation of TH+ cells in the SN. We note, however, that the therapeutic effects reported by Zhang et al. [129] were also observed in control groups that received nanobubbles without FUS. Feng, et al. [132] used FUS to facilitated the delivery of Triptolide (T_10_) into the brains of alpha-synuclein-overexpressing mice. Triptolide is a compound found in a Chinese medicinal herb that has been found to increase autophagy of alpha-synuclein in vitro [133] but does not easily cross the BBB [134]. Delivery of Triptolide with the aid of FUS promoted autophagic clearance of alpha-synuclein, attenuated the loss of TH+ cells in the SN, and resulted in an improvement of motor impairments. Another study [135] examined FUS-MB-assisted delivery of Gastrodin, a compound from another traditional Chinese medicinal herb, *Gastrodia elata*, which may offer neuroprotection through anti-oxidant, anti-inflammatory, and anti-apoptotic effects [135,136]. The FUS-MB technique was used to open the BBB in the area of the striatum to allow the entry of injected Gastrodin. This had protective effects on TH+ cells and DAT in the nigrostriatal pathway of MPTP mice as well as enhanced expression of synaptic-related proteins and anti-apoptotic activity, with some of the latter effect also evident with FUS or Gastrodin alone. However, no significant motor function improvement resulted from this treatment, as motor function auto-recovered in all lesioned animals with this sub-acute MPTP model.

*Gene Silencing*. A unique study by Xhima, et al. [137] utilized FUS-MBs to facilitate the non-invasive delivery a virally expressed short hairpin RNA (shRNA) to silence the alpha-synuclein gene in transgenic mice expressing human alpha-synuclein. Several brain areas known to be vulnerable to Lewy body pathology were targeted: SNc, hippocampus, olfactory bulb, and the dorsal motor nucleus of the vagus. Alpha-synuclein expression was decreased 1 month following treatment in mice that received FUS with active shRNA relative to mice that received FUS with scrambled shRNA. In contrast, neuronal markers, cell death, and glial activation remained unchanged following treatment [137]. This study demonstrates that the FUS-MB technique holds promise for the non-invasive delivery of brain-targeted gene therapy to curb the spread of alpha-synuclein and therefore alter disease progression.

*Creation of PD model.* Conversely, the same FUS-MB technique was employed to create a novel mouse model of PD by introducing the alpha-synuclein gene into the SN dopamine neurons [138]. The delivery of the alpha-synuclein gene accomplished a PD-like spread of alpha-synuclein with an accompanying loss of 50% of SN dopamine, which was associated with motor impairment. Although this study does not meet the inclusion criteria for the current review, we mention it as relevant to the present discussion.

*Electrical Stimulation.* Finally, one study utilized high-intensity FUS to non-invasively generate direct current in the STN by sonicating systemically administered piezoelectric nanoparticles releasing nitric oxide [139]. The release of nitric oxide disrupted the BBB, allowing for the entry of the nanoparticles into the parenchyma. Kim, et al. [139] reported that in-vitro, the piezoelectrically-induced current stimulated dopamine release from dopaminergic-like neurons. In MPTP mice, the injection of the nanoparticles coupled with the application of HIFU to the STN improved motor performance and fear learning, enhanced neural activity in the STN, and attenuated TH+ cell loss in the SN. No tissue damage was observed. This treatment was proposed to be analogous to DBS with the advantage of acting non-invasively.

In summary, studies utilizing LIFU and FUS-MBs detected no brain tissue damage, suggesting the procedure is safe at low intensities. Even without the delivery of exogenous therapeutics via FUS-MBs, studies employing variations of LIFU showed that this treatment increased the density and/ or integrity of dopaminergic SNc neurons and produced improvements in motor functioning. In addition, some of these studies produced evidence of increases in NFs [104,109,110] and reduction in inflammatory markers [108,109], oxidative stress [106,108], and apoptosis [105], pointing to some possible therapeutic mechanisms. These effects are similar to the ones produced by other electrical stimulation methods reviewed earlier. Studies of the FUS-MB-assisted delivery of therapeutics reported similar effects, including increases in dopaminergic cells, dopaminergic transmission and improvements in motor function. However, in some cases, this treatment produced an almost complete restoration of motor function and dopaminergic signaling [116,117,118,125,128,129,135], even in the 6-OHDA model. Furthermore, the study by Xhima, et al. [137] demonstrated that this approach can be used successfully to curb the spread of alpha-synuclein in brain areas that are vulnerable to Lewy body pathology, raising the possibility that this application of the FUS-MB approach has the potential to alter disease progression. However, this technique has yet to be applied in human patients.

**Table 6 jcm-12-05448-t006:** Specifications of FUS-MB studies. PRF = pulse repetition frequency; SN = substantia nigra; SNc = substantial nigra pars compacta; SNr = substantia nigra pars reticulata; OB = olfactory bulb; DMN = dorsal motor nucleus; BDNF = brain-derived neurotrophic factor; GDNF = glial-derived neurotrophic factor; pGDNF = plasmid glial cell-line-derived neurotrophic factor; α-syn = alpha-synuclein; AAV = adeno-associated viral vector; shRNA = short hairpin RNA; BPN = brain-penetrating nanoparticles; NTN = neurturin; DAT = dopamine transporter; TH = tyrosine hydroxylase; Nurr1 = nuclear receptor-related factor 1; Nrf2 = nuclear factor E2-related factor 2; ROS = reactive oxygen species; SOD = superoxide dismutase; PSp-MBs = polyethylenimine–superparamagnetic iron oxide plasmid DNA load microbubbles; PLGA = poly(lactide-co-glycolide); Bcl2 =B-cell lymphoma 2; PSD-95 = postsynaptic density protein 95.

Study	PD Model	Sham Control	FUS-MB Parameters	Outcome Measures
Fan, et al. (2016) [115]	6-OHDA (rat)	Control groups receiving GDNF, MBs, GDNF-MBs, and FUS, alone or in different combinations No sham lesioning group (untreated rats as control)	FUS + MB–GDNF gene complex delivery (gene plasmid coupled to MBs) Target: SN and striatum Frequency: 1 MHz, PRF: 1 Hz Duration: 90 s	Increased GDNF expression at target sites both in in 6-OHDA and control rats Restored GDNF concentrations in 6-OHDA rats to 85.6% of un-lesioned side Increased dopamine levels in whole brain Increased TH+ cells in SNc Increased dopamine in striatum Improved in rotational test and bar test performance
Lin, et al. (2016) [116]	MPTP (mouse)	Control groups receiving GDNF gene, MBs, and FUS, alone or in different combinations Saline injection	FUS + MBs conjugated to GDNF gene carrying liposomes Target: SN Frequency: 1 MHz, PRF: 1 Hz Duration: 60 s	Increased GDNF expression in striatum Increased levels of dopamine and its metabolites in striatum Increased (and restored to normal) TH+ neurons count in SNc Increased levels of dopamine transporter in SN and striatum Improved rotarod test performance and increased home cage activity
Long, et al. (2017) [121]	6-OHDA (rat)	Control groups receiving, no treatment; MBs + FUS w/o Nrf2 gene; Nrf2 gene alone; MBs +Nrf2 gene w/o FUS Sham lesioning	FUS + MBs (nanomicrobubbles) containing Nrf2 gene Target: SN Frequency:0.5 MHz Duration: 30 s	Increased Nrf2 expression in SN Increased TH and DAT levels in SN Decreased ROS in SN Increased SOD in SN
Mead, et al. (2017) [117]	6-OHDA (rat)	Control groups receiving FUS + sham BPN, GDNF-BPN without FUS No sham lesioning or untreated group	FUS-MBs + brain-penetrating nanoparticles (BPNs) carrying GDNF gene Target: striatum Frequency: 1.15 MHz, PRF: 0.5 Hz Duration: 2 min	Increased GDNF expression in striatum Increase in dopamine and dopamine metabolite levels in striatum to nearly un-lesioned side levels Increased TH+ neuron density in SNc and striatum Improved rotational behavior and reduced forepaw use bias to pre-lesion levels No damage to brain tissue
Xhima, et al. (2018) [137]	Transgenic humanized alpha-synuclein (mouse)	FUS-MBs + AAV with scrambled shRNA; AAV with scrambled shRNA w/o FUS	FUS-MBs + AAV with α-syn silencing shRNA Targets: SN, hippocampus, OB, DMN Frequency: 1.68 MHz, PRF: 1 Hz Duration: 120 s	Reduction in α-syn in all target areas 1 month following treatment No evidence of cell death or immune activation following treatment
Yue, et al. (2018) [118]	6-OHDA (rat)	Controls groups receiving vehicle injection without FUS; PEGylated liposomes without GDNF + MBs + FUS Sham lesioning group	FUS + MBs conjugated to GDNF gene carrying PEGylated liposomes Target: SN Frequency: 1 MHz Duration: unspecified	Increased expression of GDNF and Nurr1 in SN Increased TH and DAT levels in SN Improved rotational behavior test Improved suspension test performance Improved pole test performance
Zhang, et al. (2018) [129] **	MPTP (mouse)	Groups receiving CPC with no PS 80 and no FUS, CPC with PS 80 and no FUS, PS 80 modified cerasomes with no curcumin + FUS, CPC with PS 80 + FUS No sham lesioning group (healthy mice used as controls)	FUS + injection of polysorbate 80 (PS 80)-modified cerasomes containing curcumin (CPC) Target: striatum Frequency: 1.28 MHz PRF: unspecified Duration: 60 s, every 2 days, total of 4 times	Improved rotarod test performance Improved pole climbing test performance Behavioral improvements observed also in mice that received CPC with PS 80 but no FUS Increased dopamine and dopamine metabolites in striatum Increased TH+ staining in SN Dopamine, metabolite, and TH+ increases also observed in control groups No evidence of histological damage
Niu, et al. (2018) [125] **	6-OHDA (rat)	No sham stimulation group; control groups receiving FGF20 or FGF20 in liposomes without FUS. Sham lesioned group	FUS + rhFGF20 proteoliposomes Target: region unspecified Frequency: 0.69 MHz PRF: I Hz Duration: 60 s every day for 2 weeks	Improvements in rotational behavior test No tissue damage Alleviation of TH+ cell loss in SN
Ji, et al. (2019) [123]	MPTP (mouse)	FUS-MBs w/o BDNF; no treatment No sham lesioning	FUS-MBs + intranasal administration of BDNF Target: left basal ganglia (SN and striatum) Frequency: 1.5 MHz, PRF: 10 Hz Duration: 60 s per region, once a week for 3 weeks	Increased TH in striatum and SN Emergence of rotational behavior bias towards the untreated side (indicating increased dopaminergic activity on treated side)
Karakatsani, et al. (2019) [119]	MPTP (mouse, early PD model)	Control groups receiving NTN, GDNF, and FUS alone No sham lesioning group (untreated rats as control)	FUS-MBs + injection of NTN (protein) or GDNF (gene) Targets: SN and caudate-putamen Frequency: 1.5 MHz, PRF: 10 Hz Duration: either 1 or 3 treatments, lasting 60 s	Increased count of TH+ neurons in SCc and density of dopaminergic fibers in SNr and terminals in caudate-putamen following NTN + FUS-MBs and AAV-GDNF + FUS-MBs Increased dopamine levels in ventral midbrain following NTN + FUS-MBs Effects of NTN more pronounced—and some only observed—after repeated protein delivery Emergence of rotational behavior bias towards the untreated side following AAV-GDNF + FUS-MBs (indicating increased dopaminergic activity on treated side)
Wu, et al. (2020) [122] **	MitoPark (mouse, genetic)	Control groups receiving pGDNF + FUS+ MBs without PSp and without magnetic navigation (MN), PSp-MBs + FUS without MN, and PSp-MBs + FUS + 1st step MN Control group of healthy mice	PSp-MBs + FUS + two-step magnetic navigation Target: SN Frequency: 1 MHz, PRF: 1 Hz Duration: 1 min	Improvement in balance Increased distance traveled in open field Increased GDNF expression in SN Reduced loss of TH+ cells in sonicated SN
Yan, et al. (2021) [128]	MPTP (mouse)	Control groups receiving LIFU alone and Cur-NBs alone No sham lesioning group (healthy mice as control group)	FUS + injection of curcumin-loaded lipid-PLGA nanobubbles (Cur-NBs) Target: striatum Frequency: 1 MHz PRF: 1 Hz Duration: 1 min, once every other day, 6 times	Improvement performance on rotarod Improvement on pole climbing test
Feng, et al. (2022) [132] **	rAAV2/5-wild type α-syn or A53T α-syn (mouse)	Control groups receiving T_10_ alone, FUS alone, T_10_ + MBs administered separately + FUS, T_10_ + MB complex + FUS (without AHNAK targeting) Sham of α-syn viral injection	FUS + injection of AHNAK-targeted MBs containing triptolide (T_10_) Target: right hemisphere Frequency: 620 kHz PRF: 1 Hz Duration: 60 s, twice a week for 3 weeks	Enhanced delivery of Triptolide into the brain with AHNAK-targeted MBs Improvement rotarod performance Reduced loss of TH+ cells in SN Increased autophagic clearance of α-syn
Wang, et al. (2022) [135]	MPTP (mouse)	Control groups receiving Gastrodin without FUS-MBs and FUS-MBs without Gastrodin Sham lesioning group	FUS-MBs + injection Gastrodin Target: striatum Frequency: 1 MHz PRF: 1 Hz Duration: 60 s, once every 3 days, total of 6 times	Auto-recovery of motor performance on pole and pole grip endurance tests in all groups; no significant effect of FUS-MBs + Gastrodin on motor performance Increased TH+ and DAT in nigrostriatal pathway Decreased amount of cleaved-caspace-3 in striatum Increased Bcl2 in striatum Increased levels of BDNF, synaptophysin, and PSD-95 in striatum
Kim, et al. (2023) [139]	MPTP (mouse)	Control groups treated with saline, nanoparticles + 1 FUS application, no nanoparticles + multiple FUS applications, and non-piezoelectric nanoparticles + multiple FUS applications Sham lesioning group	Injected piezoelectric nanoparticles + high-intensity FUS Target: STN Frequency: 1.5 MHz PRF: 10 Hz Duration: 60 s for 16 days	Improvement in rotarod performance Increased distance traveled in open field Enhanced freezing in fear memory recall test Enhanced c-fos staining in STN Attenuation of TH+ cell loss in SN No tissue damage

** This study also performed in-vitro investigations; results not discussed in this review.

#### 3.4.4. Limitations

Although findings regarding the effects of FUS in animal models of PD have been promising, several limitations must be noted regarding its potential for use in human patients. As with the tDCS studies, the FUS delivery protocols have differed substantially across studies. With the FUS-MB technique, the risk of inflammation resulting from the disruption of the BBB in the targeted areas presents a concern [140]. However, the studies reviewed herein did not report evidence of inflammation, and some studies in other animal models have suggested that inflammation may be transient [141,142,143]. On the other hand, LIFU was found to help restore the integrity of the BBB [108], although more studies are needed to replicate this finding. As noted for the other forms of neuromodulation reviewed here, possible interactions of FUS with dopamine replacement therapy remain to be studied, and its effects on non-motor symptoms of PD remain to be investigated. In addition, despite its superior spatial resolution of FUS, it may be less anatomically precise in small animal models considering the small brain size of these animals. Finally, the optimal frequency and duration for sonication of human patients remains to be determined.

## 4. Discussion and Future Directions

Animal models have offered important insights regarding possible therapeutic mechanisms of non-invasive neuromodulation treatments for PD. Our literature review has identified some therapeutic endpoints and mechanisms that have remained elusive or understudied in human patients (Figure 2).

Specifically, the capacity of most neuromodulation methods reviewed here (TMS, tDCS/tACS, ECT, and FUS) to influence neuroinflammation deserves more attention in human patient research. PET tracers exist for imaging neuroinflammation [47,144]. While neuroinflammation imaging in PD is novel [145], evidence is accumulating for the role of neuroinflammation in PD pathophysiology [44,146]. Leveraging PET imaging to observe changes in neuroinflammation in response to non-invasive neuromodulation treatments may be a promising future direction. However, the PET imaging of neuroinflammation comes with methodological challenges, such as low binding specificity of tracers to their targets, variable tracer kinetics, and genetic variation in translocator protein, determining the affinity to TSPO tracers for this protein in activated microglia [47,147]. However, improvements in radiotracer signal to noise ratio, kinetics, and binding affinity will decrease these limitations in the future [47].

Another prominent set of findings that emerged from the studies reviewed here are neuromodulation-induced increases in NFs. This is a valuable mechanistic insight made possible uniquely by research in animal models. Furthermore, several studies have employed FUS to aid in the delivery of NFs to brain areas affected by PD pathology as therapeutics with encouraging results. This presents an intriguing avenue for future research in PD patients. However, a major limitation in this research is that NFs are not currently measurable in the brains of human patients in-vivo. While it is possible to quantify BDNF in the body through blood serum analysis and in the cerebrospinal fluid, these values do not directly reflect levels in the PD-related brain regions. Although it will not be possible to measure changes in the levels of NFs in human PD patients as a result of interventions such as FUS-assisted NF delivery, therapeutic effects of such interventions may be assessed as behavioral changes, as well as PET measures of pre- and postsynaptic dopamine function. This major limitation will make it difficult to compare NF levels between the human and animal literature as the majority of the animal literature quantifies NFs in the brain post-mortem.

TMS research in animal models of PD is limited by the low spatial resolution in small animals. rTMS studies in non-human primate models may be more informative regarding regional effects, as they would allow for more focal applications. The existing TMS animal literature suggests that rTMS improves motor function, restores synaptic plasticity, enhances the survival of SN dopamine neurons, and may increase NFs and reduce neuroinflammation in models of PD. However, these studies utilized a lesioning model of PD, and lesioning itself produces inflammation and oxidative stress. The replication of these findings using a transgenic PD model would enhance the translational validity of these findings.

tDCS studies in animal models of PD showed similar effects to those of rTMS; namely tDCS resulted in the improvement of motor function, as well as increases in dopamine cells and anti-neuroinflammatory effects. Notably, findings regarding the effects of tDCS on neuropsychiatric symptoms associated with PD were scarce and mixed, with one study reporting improvements in anxiety-like but not in depressive-like behaviors [71]. The effects of tDCS on non-motor symptoms require further study. Unlike TMS, tDCS has reached the non-human primate literature and has shown efficacy in decreasing tremor [77]. This study was in an advanced model of PD, suggesting that tDCS may still be effective even at later stages of the disease. Overall, considering the relative ease of use and accessibility of tDCS combined with its capacity to produce similar therapeutic effects to those of rTMS at behavioral, cellular, and molecular levels, tDCS is poised to become a helpful and accessible adjunctive treatment for PD patients.

ECT studies in animal models of PD were few, and their translatability to human patients was limited by the absence of anesthesia accompanying the procedure in most of the reviewed animal studies. The reported effects were similar to the ones observed with TMS and tDCS, namely the improvement of motor deficits, the preservation of dopamine neurons, and increases in NFs. Intriguingly, one study [88] found evidence that ECT may impact autophagy pathways in PD, although it is unclear whether this finding would translate to human patients. Considering that other forms of non-invasive neuromodulations are likely to offer similar benefits based on the studies reviewed here, while being associated with fewer risks, ECT may be less preferable in clinical practice than other forms of non-invasive neuromodulation.

FUS is the neuromodulation technique that has been most studied in animal models of PD. While some of these studies employed FUS to achieve the focal modulation of neural activity in the areas affected by PD pathology, others used this technique in combination with injected microbubbles to disrupt the BBB and aid the delivery of therapeutics. The former group of studies reported effects that were similar to those of the other neuromodulation techniques reviewed, including improvements in motor performance, the improved survival of dopamine neurons, increases in neural activity in the targeted circuits, decreased neuroinflammation, antioxidant effects, and increases in NFs along with facilitation of structural plasticity. The group of studies using FUS to enable the delivery of NFs or gene therapy have demonstrated effective delivery of these therapeutics to focal brain regions with beneficial (and sometimes fully restorative) effects on motor function and key aspects of PD pathophysiology. This application of FUS for the focal delivery of therapeutics is a unique capability of this technique, which could open avenues for ground-breaking and even disease-altering treatments for PD. Another the clear advantage of FUS is its superior spatial resolution and capacity to precisely target deep as well as cortical brain areas. This allows for a more precise and targeted modulation of dysfunctional circuits. However, the relative novelty of this technique and the paucity of studies in human patients present barriers to its imminent widespread use in clinical practice.

Many of the studies reviewed focused on motor symptoms as an outcome, which does not represent the full spectrum of PD symptoms. Cognitive and neuropsychiatric symptoms are important parts of the pathology and a worthwhile focus for future studies in animal models of PD. However, investigation into neuropsychiatric symptoms in animals is more challenging than studying the motor signs, with some questioning the validity of tests, such as the forced swim test used to assess states like depression in rodents [148]. For example, the forced swim test may actually be measuring a rodent’s motivation rather than despair [148]. Administering multiple tests assessing non-motor symptoms as a battery may help increase confidence in the measurement of cognitive phenotypes and boost translational validity [149]. In addition, certain non-motor symptoms such as feelings of dizziness or orthostasis are not measurable in animal models.

In addition, the motor symptoms investigated in the studies reviewed here over-represented rotational behavior as an outcome. Fewer studies utilized other tests like gait analysis, open field locomotion, or tests of motor coordination. Although the rotational behavior test is considered the gold standard, especially for unilateral lesion models (6-OHDA), other tests would give a more complete picture of the alleviation of motor symptoms.

Most of the studies reviewed utilized lesioning models of PD, most commonly the unilateral 6-OHDA lesioning model in rats and the bilateral MPTP lesioning model in mice. Lesioning models come with a set of limitations and concerns (e.g., surgery risks, inflammation, and stress on the animals). First, lesioned animals have a high mortality rate (6-OHDA model = 16% [150,151], chronic MPTP model~15% [152] mortality rate), leading to their use to acutely model a rapid onset of severe PD, instead of the chronic, progressive condition that human patients face. Second, both the 6-OHDA and MPTP models lack Lewy body inclusions, which is the cardinal feature of PD. Transgenic animal models of PD would avoid some of these complications and allow for better evaluations of these neuromodulation methods in aged animals (e.g., parkin null mice can live up to 24 months [153]). However, transgenic animals model genetic forms of PD, not idiopathic PD, which accounts for the majority of PD cases. Idiopathic PD and genetic forms of the disease differ in their molecular mechanisms [154].

Differences between the 6-OHDA and MPTP models must be noted when considering findings across studies employing the two models. 6-OHDA does not cross the BBB and needs to be administered intracerebrally. It enters the dopamine cells through the dopamine transporter and oxidizes rapidly producing reactive oxygen species and mitochondrial dysfunction causing the degeneration of neurons [155]. Usually, 6-OHDA is injected in the SN, medial forebrain bundle, or striatum [151]. The majority of 6-OHDA studies reviewed here (i.e., 12 studies total) injected 6-OHDA in the medial forebrain bundle, which is known to cause the long-lasting relatively slow degeneration of SN neurons over up to 5 weeks [156]. Seven studies injected 6-OHDA into the striatum, which is known to cause the rapid degeneration of local terminals, followed by a more gradual degeneration of nigral cell bodies (for up to several weeks), which is believed to more accurately reflect the pathophysiology of PD [151,157]. The lesioning site could affect response to the non-invasive brain stimulation. MPTP is a lipophilic molecule that can be administered systemically. It is oxidized into a dopaminergic neurotoxin 1-methyl-4-phenylpyridinium ion (MPP+), which enters the cell through the dopamine transporter and causes a progressive loss of dopamine neurons. Unlike the 6-OHDA model, the MPTP model requires repeated lesioning [155]. Thus, in the studies using the MPTP model, the neuromodulation treatments sometimes occurred concurrently with the lesioning [70,72,82,87,108,116,128,129], in which case the prevention rather than the remediation of damage was assessed. In the studies using the 6-OHDA model, treatments occurred following the lesioning and produced remediation rather than prevention of damage. In unilaterally lesioned animals (mainly 6-OHDA) comparisons were typically made between the lesioned and the un-lesioned side for both behavioral and neural effects, whereas in bilaterally lesioned animals, comparisons with sham-lesioned animals were typically employed. Finally, different behavioral tests were used to assess motor function depending on the model. As mentioned earlier, the rotational behavior test predominated in the studies employing unilateral lesioning (mainly 6-OHDA) models, whereas bilateral lesioning models (such as MPTP) more often used the rotarod test to test motor functions. Despite these differences, the findings pertaining to the neurobiological effects of neuromodulation were qualitatively similar between the 6-OHDA and MPTP model studies, including improvement in motor function, the attenuation of dopaminergic cell loss in the SN, increases in NFs in key brain regions, and decreases in pro-inflammatory markers in PD-related brain regions.

Finally, the results of the studies reviewed do not speak to whether/how the different neuromodulation methods may be combined with each other or with other PD treatments. We note two studies that combined stem cell transplantation with tDCS [76] and ECT [87], showing benefits from the adjunctive neuromodulation. Evaluating how different forms of non-invasive neuromodulation may be combined with more conventional or breakthrough treatments is an important future direction.

### 4.1. Applications to Clinical Practice

Although the animal studies reviewed here have produced important insights into the therapeutic mechanisms of the brain modulation techniques, they provide little information that is pertinent to the application of these techniques in clinical practice. For a discussion of clinical applications of these brain modulation techniques in PD, we refer the reader to the following reviews: [14,96,158,159]. The literature in animal models aligns with the human patient literature in that the majority of studies have focused on and provided evidence for the amelioration of motor symptoms as the primary outcome. Hence, the literature primarily supports the clinical use of these techniques for the treatment of the motor symptoms, although this may be an artifact of motor symptoms being most commonly used as the primary outcome. More studies of these methods for non-motor symptoms are needed to inform clinical practice. Considering that non-motor symptoms can be the most debilitating aspect of the disease [160], this is an important future direction.

Besides motor function improvement, the studies in animal models have revealed the neuroprotective/neuroregenerative effects of the brain modulation methods reviewed here. To the degree that these effects may translate to human PD patients, they may be disease-modifying. Given the challenges of demonstrating neuroprotective or anti-neuroinflammatory effects in human patients, such findings in animal models alone may provide some rationale for prophylactic use of these techniques, especially considering their rather benign side effect profile. However, the duration of the neuroprotective effects following treatment discontinuation remains unknown. Perhaps with the exception of FUS-assisted gene therapy, the effects are likely to be transient. Importantly, the effects of long-term chronic administration of these forms of neuromodulation (as would be done for prophylaxis) are also unknown.

Qualitatively similar therapeutic effects were observed across all forms of neuromodulation reviewed at both the behavioral and neurobiological levels. As noted in the Limitations subsection, this review did not attempt to quantify effect sizes and compare them across the different brain modulation approaches. We are, therefore, unable to compare the therapeutic efficacy of the different neuromodulation techniques. Relative to the other methods, the FUS-MB technique has had the clear advantage of enabling the delivery of therapeutics to the brain and hence shows the most promise as a disease-modifying therapy. In addition, some of the studies reviewed here reported therapeutic benefits following a single treatment with FUS-MBs or LIFU, whereas repeated treatments with rTMS, tDCS, and ECT were employed by studies demonstrating more than transient therapeutic effects. On the other hand, FUS-MBs is the least studied (in humans) of the methods reviewed and has not yet been adopted into clinical practice, although the initial studies demonstrate that this technique is well-tolerated and can be safely used in humans [99,100] which is in agreement with the animal studies reporting no tissue damage.

Besides efficacy, important considerations for clinical applications are safety, tolerability, and accessibility. rTMS, tDCS/ tACS, and ECT all have favorable safety profiles. The delivery of rTMS, tDCS, and tACS is generally painless. The most common side effects of these treatments are headaches [161,162,163], and patients also report itching, tingling or burning sensations on the scalp with tDCS and tACS [162]. In addition, in patients with a history of seizure disorders, rTMS may induce seizures, which are otherwise a rare side effect of rTMS [164]. Therefore, this treatment is not suitable for those with a history of seizures. ECT has the least favorable side effect profile as it may produce transient cognitive side effects, including confusion and delirium (in addition to headaches and nausea being common side effects) [161]. Hence, this treatment is less appropriate for patients already experiencing cognitive difficulties or lacking strong family or social support systems. Regarding accessibility, tDCS is poised to become the most accessible form of neuromodulation. Although not currently FDA-approved for any condition, this treatment has regulatory approval in the European Union and other countries, with some devices approved for home use. One such device has been recently granted an FDA Breakthrough designation. The devices are lightweight, portable, easy to use, and relatively inexpensive. rTMS and ECT both necessitate frequent visits to treatment facilities. Although ECT requires less frequent sessions and fewer total treatments than TMS, it requires short-acting general anesthesia, which may be undesirable or contraindicated for some patients because of other medical conditions.

In addition, some PD patients may already be undergoing DBS, and there is minimal literature that speaks to whether the non-invasive brain modulation methods reviewed are compatible with DBS. In a brain phantom model of PD, researchers found that combining DBS and TMS will cause overstimulation [165]. A case report has found that two patients who had undergone DBS did not show any substantial side effects when receiving tDCS [166]. Similarly, several case reports investigating use of ECT on PD patients with a DBS stimulator (stimulator off during ECT) did not show any adverse effects, but showed an amelioration of depressive symptoms [167,168,169,170,171]. For FUS combined with DBS, one study using a brain phantom did not find any dangerous increases in brain tissue temperature that would pose a safety concern in patients that have a DBS stimulator [172].

Finally, we note that the lesions induced by neurotoxins to produce the PD models for the majority of the studies we reviewed are considered severe and thought to mimic late-stage neurodegeneration (albeit in young animals) [173]. Therefore, the therapeutic mechanisms reported by these studies may be more pertinent for late-stage PD, although studies in human patients are required to determine this definitively.

### 4.2. Limitations

This is not a systematic review but rather a rapid review, whose goal was to survey the literature. As such, we employed a less exhaustive search strategy than would be expected of a systematic review. Crucially, we did not perform a risk of bias or quality assessment of the included studies and hence cannot make judgements about the quality of the articles reviewed; although, we make note of methodological issues, such as the absence of controls. Furthermore, we did not calculate the effect sizes in the included studies. Therefore, we cannot compare the effects across the different non-invasive brain modulation methods and evaluate their relative efficacy for specific outcomes. A meta-analysis or a meta-analytic review would be a valuable future addition to the literature. Additionally, we acknowledge that we have not covered every non-invasive method currently available. We have focused on the most promising brain modulation techniques in the human PD literature that have been studied in animal models of PD. Therefore, we have not discussed methods such as electro-acupuncture or transcutaneous nerve stimulation.

### 4.3. Conclusions

In conclusion, the literature reviewed herein has supported findings from human patient literature pointing to the clinical utility of non-invasive neuromodulation techniques as treatments for PD. Crucially, it has identified cellular and molecular mechanisms underlying the therapeutic effects of these techniques and have pointed to promising future directions for research and treatment in human patients. Further research is necessary to determine whether these therapeutic mechanisms translate to human PD patients and how they can best be leveraged to produce maximal therapeutic benefits.

## Figures and Tables

**Figure 1 jcm-12-05448-f001:**
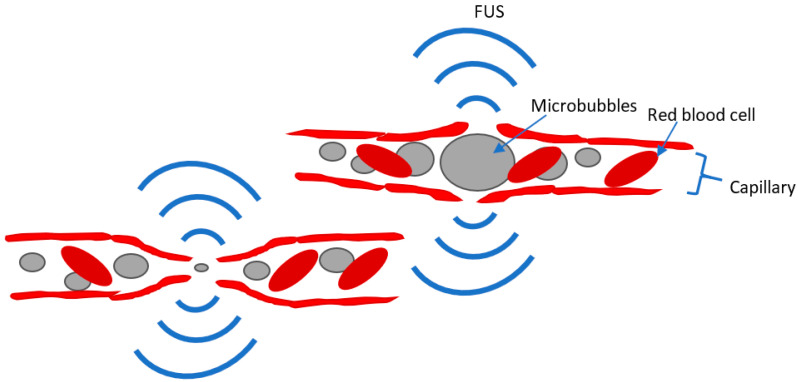
Depiction of FUS-MB mechanism for BBB opening. MBs expand and contract within the capillary in response to FUS and cause a brief disruption of the BBB, which allows systemically injected substances to enter through the BBB. Adapted from Wu, et al. [20].

**Figure 2 jcm-12-05448-f002:**
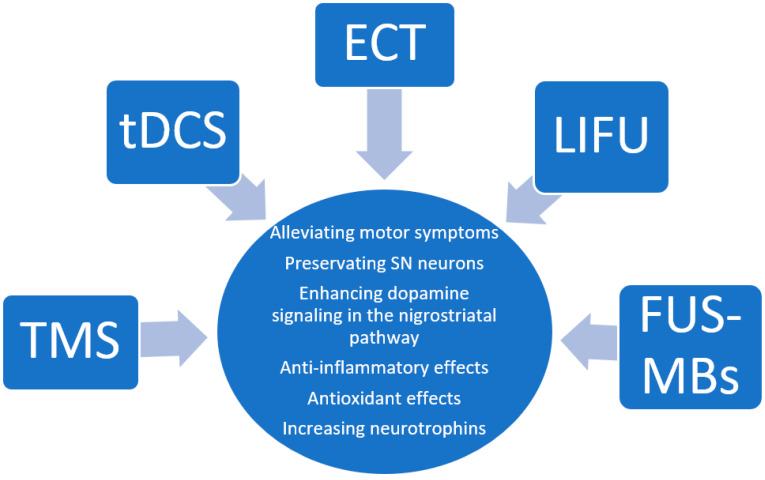
Insights gathered from the animal literature regarding the therapeutic mechanisms of the neuromodulation techniques reviewed.

**Table 1 jcm-12-05448-t001:** Summary of evidence from reviews and meta-analyses on alleviation of PD-related symptoms in humans. Hf-rTMS = high-frequency rTMS; lf-rTMS = low-frequency rTMS; cTBS = continuous theta burst stimulation; iTBS = intermittent theta burst stimulation; M1 = primary motor cortex; SMA = supplementary motor area; DLPFC = dorsolateral prefrontal cortex.

TMS Type	Stimulated Regions	Primary Symptom Relief
hf-rTMS	Frontal, prefrontal, M1, DLPFC, SMA, inferior frontal gyrus, dorsal premotor cortex, occipital cortex	Motor signs [12,13,14,15,16,17] Cognitive symptoms [14,18] Depression [17]
lf-rTMS	Vertex, frontal, prefrontal, M1, DLPFC, SMA	Motor signs [17,19] Levodopa-induced dyskinesia [20]
cTBS	SMA	Motor signs in the unmedicated state [9]
iTBS	M1, DLPFC	Slowing of gait in the unmedicated state [9] Depression [9]

## Data Availability

No new data were created or analyzed in this study. Data sharing is not applicable to this article.

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
