# Peer review of "Noninvasive Neuromodulation in Parkinson’s Disease: Insights from Animal Models"

_jcm, 2023, doi:10.3390/jcm12175448_

Round 1

Reviewer 1 Report

This review article discussed the ongoing research into non-invasive neuromodulation methods (like TMS, tDCS, ECT, and FUS) as potential treatments for Parkinson's Disease. It also mentions studies on animal models to understand the possible therapeutic mechanisms of these methods. For each of the non-invasive neuromodulation methos, the discuss lies on the background, animal studies, and limitations, which allows readers to understand these treatment methods at both a macro and detailed level. But I still have some suggestions for the authors as following:

1.     As I know, except the mentioned neuromodulation methods, there are some other existing methods was not included in this paper, like the: Alternating Current Stimulation, and Transcutaneous Nerve Stimulation. I hope the authors can discuss these two methods.

2.     In the discussion part, the discussion highlights unresolved issues and areas that require further investigation, such as neuroinflammation, neurotrophic factors, and non-motor symptoms. However, it does not provide a detailed exploration of the specific limitations and challenges associated with these issues. For instance, studying the impact of neuroinflammation in human research may face methodological difficulties and technological constraints. Therefore, further discussion is needed to address these issues and identify viable methods for gaining a deeper understanding of these mechanisms.

3.     Also, the authors mentioned kinds of treatment methods for the Parkinson's Disease. It is important to note that the effectiveness and scope of these methods may vary due to individual differences. For each patient, the selection of appropriate non-invasive neuromodulation therapy should be determined by medical professionals based on specific conditions and clinical evaluations. The author can discuss the relative applicability of these methods for different issues. When discussing the relative applicability of these methods, the author can focus on the following aspects:

1)     Different symptoms of Parkinson's disease: Discuss which methods are more effective in alleviating motor impairments, non-motor symptoms, or other specific symptoms.

2)     Duration of therapeutic effects: Discuss the duration of the effects of these methods after treatment to assess the need for repeat therapy.

3)     Patient's physical condition: Discuss whether there are specific patient groups, such as age, disease progression, severity of symptoms, etc., that may benefit more from certain methods.

4)     Safety and side effects: Discuss the safety and potential side effects of different methods to ensure that the patient's health is fully considered when selecting the treatment method.

Through such discussions, readers will gain a better understanding of which non-invasive neuromodulation therapies are more advantageous in addressing specific issues related to Parkinson's disease, thereby providing a more comprehensive reference and guidance for clinical practice.

Reviewer 2 Report

In this paper, Katherine et al. summarized the noninvasive neuromodulation in parkinson’s disease animal models. It is suggested to improve relevant research contents.

1.     In the materials and methods section, it is recommended to supplement the retrieval strategy for "animal models" with the following terms also included: TMS –“AND transcranial magnetic stimulation”; tDCS – “AND transcranial direct current stimulation”; ECT – “AND electroconvulsive therapy”; FUS – “AND focused ultrasound” as well as “AND transcranial ultrasound”.

2.     When describing various animal models, it is recommended that the author categorize the content according to a unified standard, and then describe it according to different categories to increase the readability of the paper.

For example:

3.1.2

Classification standard 1: Different treatment methods (Rtms,iTBS)

Or Classification standard 2: Different mechanisms of action (restoring deficient synaptic plasticity, anti-inflammatory effects, etc.)

3.     In the discussion section, can the author discuss the effects and mechanisms of different treatment methods on PD animal models generated using the same modeling method, so as to have more comparability and reference significance.

4.     Due to the different mechanisms and effects of different treatment methods, based on the results of existing animal models, can different treatment methods be combined to compensate for the shortcomings between different treatment methods and achieve the optimal treatment effect? It is recommended that the author supplement relevant discussions.

Reviewer 3 Report

Please, delete the term "rapid review" from paper.

I think it is a very interesting research but you have to improve it following PRISMA check list for systematic reviews, including a flow chart of study selection, explaing the peer review process to select them, include risk of bias and quality assesment of articles evaluation accodrding to articles design

Round 2

Reviewer 1 Report

All concerns and recommendations have been comprehensively addressed. The quality of the paper has seen a remarkable improvement. I am confident that this paper will provide readers with an insightful perspective on the recent advancements in this field.

Reviewer 2 Report

none